# Wavelet Networks: Scale-Translation Equivariant Learning From Raw Time-Series

**David W. Romero**[*]                                            *dwromero@nvidia.com*
*NVIDIA Research*

**Erik J. Bekkers**                                               *e.j.bekkers@uva.nl*
*Universiteit van Amsterdam*

**Jakub M. Tomczak**[*]                                           *j.m.tomczak@tue.nl*
*Technische Universiteit Eindhoven*

**Mark Hoogendoorn**                                             *m.hoogendoorn@vu.nl*
*Vrije Universiteit Amsterdam*

**Reviewed on OpenReview:** *https: // openreview. net/ forum? id= ga5SNulYet*

## Abstract

Leveraging the symmetries inherent to specific data domains for the construction of equivariant neural networks has lead to remarkable improvements in terms of data efficiency and generalization. However, most existing research focuses on symmetries arising from planar and volumetric data, leaving a crucial data source largely underexplored: *time-series*. In this work, we fill this gap by leveraging the symmetries inherent to time-series for the construction of equivariant neural network. We identify two core symmetries: *scale and translation*, and construct scale-translation equivariant neural networks for time-series learning. Intriguingly, we find that scale-translation equivariant mappings share strong resemblance with the *wavelet transform*. Inspired by this resemblance, we term our networks *Wavelet Networks*, and show that they perform nested non-linear wavelet-like time-frequency transforms. Empirical results show that Wavelet Networks outperform conventional CNNs on raw waveforms, and match strongly engineered spectrogram techniques across several tasks and time-series types, including audio, environmental sounds, and electrical signals. Our code is publicly available at https://github.com/dwromero/wavelet_networks.

## 1 Introduction

Leveraging the symmetries inherent to specific data domains for the construction of statistical models, such as neural networks, has proven highly advantageous, by restricting the model to the family of functions that accurately describes the data. A prime example or this principle is Convolutional Neural Networks (CNNs) (LeCun et al., 1989). CNNs embrace the translation symmetries in visual data by restricting their mappings to a *convolutional* structure. Convolutions possess a distinctive property called *translation equivariance*: if the input is translated, the output undergoes an equal translation. This property endows CNNs with better data efficiency and generalization than unconstrained models like multi-layered perceptrons.

Group equivariant convolutional neural networks (G-CNNs) (Cohen & Welling, 2016) extend equivariance to more general symmetry groups through the use of *group convolutions*. Group convolutions are *group equivariant*: if the input is transformed by the symmetries described by the group, e.g., scaling, the output undergoes an equal transformation. Equivariance to larger symmetry groups endows G-CNNs with increased data efficiency and generalization on data exhibiting these symmetries. Existing group equivariance research primarily focuses on symmetries found in visual data, e.g., planar rotations, planar scaling (Weiler et al., 2018; Worrall & Welling, 2019; Sosnovik et al., 2020), and more recently, on 3D symmetries, e.g., for spherical

---

[*]Work done while at the Vrije Universiteit Amsterdam

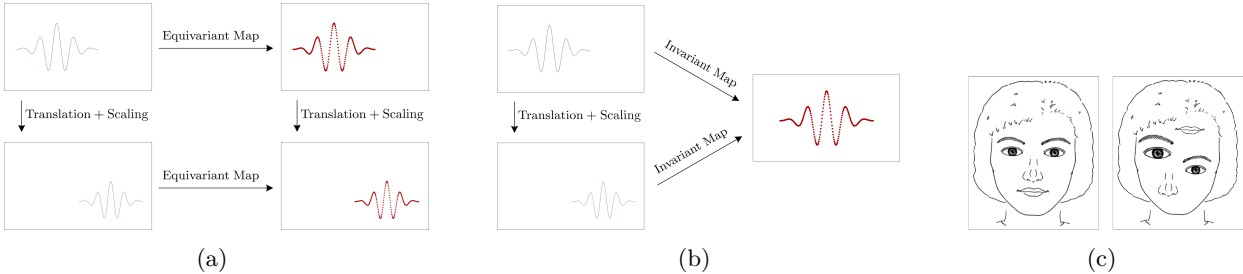

Figure 1: Equivariance, invariance and their impact on the hierarchical representations. In a group equivariant mapping, when the input is transformed by a group transformation, its output undergoes an equivalent transformation (Fig. 1a). In contrast, in group invariant maps, the output remains unchanged for all group transformations of the input (Fig. 1b). This distinction holds significant implications in the construction of hierarchical feature representations. For example, a face recognition system built upon invariant eye, nose and mouth detectors would be unable to set the portraits in Fig. 1c apart. However, by leveraging equivariant mappings, information about the input transformations can be used to distinguish these portraits effectively. In essence, in contrast to equivariant maps, invariant maps permit senseless pattern combinations resulting for overly restraining constraints in their design.

and molecular data (Thomas et al., 2018; Fuchs et al., 2020; Satorras et al., 2021). Yet, an important category remains underexplored, which also exhibits symmetries: *time-series*. Notably, their translation symmetry is a cornerstone in signal processing and system analysis, e.g., Linear Time-Invariant (LTI) systems.

In this work, we bridge this gap by constructing neural networks that embrace the symmetries inherent to time-series. We begin by asking: "*What symmetries are inherently present in time-series?*" We identify two fundamental symmetries –*scale and translation*–, whose combination elucidate several phenomena observed in time-series, e.g., temporal translations, phase shifts, temporal scaling, resolution changes, pitch shifts, seasonal occurrences, etc. By leveraging group convolutions equivariant to the *scale-translation* group, we construct neural architectures such that when the input undergoes translation, scaling or a combination of the two, all intermediate layers will undergo an equal transformation in a hierarchical manner, akin to the methods proposed by Sosnovik et al. (2020); Zhu et al. (2022) for visual data. Interestingly, we observe that constructing convolutional layers equivariant to scale and translation results in layers that closely resemble the *wavelet transform*. However, we find that in order to preserve these symmetries consistently across the whole network, the output of each layer must be processed by a layer that also behaves like the wavelet transform. This approach substantially deviates from common approaches that rely on spectro-temporal representations, e.g., the wavelet transform, which compute spectro-temporal representations once and pass their response to a 2D CNN for further processing.

Inspired by the resemblance of scale-translation group equivariant convolutions with the wavelet transform, we term our scale-translation equivariant networks for time-series processing *Wavelet Networks*. Extensive empirical results show that Wavelet Networks consistently outperform conventional CNNs operating on raw waveforms, and match strongly engineered spectogram-based approaches, e.g., on Mel-spectrograms, across several tasks and time-series types, e.g., audio, environmental sounds, electrical signals. To our best knowledge, we are first to propose scale-translation equivariant neural networks for time-series processing.

## 2 Related Work

**Learning from raw time-series.** Several end-to-end learning approaches for time-series exist (Dieleman & Schrauwen, 2014; Dieleman et al., 2016; Dai et al., 2017; Rethage et al., 2018; Stoller et al., 2018). Given the considerable high-dimensionality of time-series, existing works focus on devising techniques with parameter- and compute-efficient large memory horizons (Romero et al., 2021; Goel et al., 2022). Due to small effective memory horizons and long training times, Recurrent Neural Networks (RNNs) (Rumelhart et al., 1985) have gradually been overshadowed by CNN backbones (Bai et al., 2018).

While CNNs are equivariant to translations, they do not inherently incorporate a distinct notion of scale. Although methods involving layer-wise multi-scale representations have been proposed, e.g., Zhu et al. (2016); Lu et al. (2019); von Platen et al. (2019); Guizzo et al. (2020), these layers are not scale equivariant. As a result, networks incorporating them struggle to maintain consistent scale information across layers.

**Group-invariant time-series learning.** Learning invariant representations from raw speech and sound has been extensively studied in past. Scattering operators (Mallat, 2012; Bruna & Mallat, 2013) construct group *invariant* feature representations that can be used to construct neural architectures invariant to scale and translation (Andén & Mallat, 2014; Peddinti et al., 2014; Salamon & Bello, 2015). In contrast to the *invariant* feature representations developed by these works, Wavelet networks construct *equivariant* feature representations. Since group equivariance is a generalization of group invariance (Fig. 1b, Sec. 3.1), Wavelet Networks accommodate a broader functional family than previous works, while still upholding scale and translation preservation. Notably, equivariant methods shown superior performance compared to invariant methods across several tasks, even for intrinsically invariant tasks like classification (Cohen & Welling, 2016; Zaheer et al., 2017; Maron et al., 2018). This phenomenon stems from the hierarchical form in which neural networks extract features. Enforcing invariance early in the feature extraction process imposes an overly restrictive constraint in the resulting models (Fig. 1c).

**Group-equivariant time-series learning.** To our best knowledge, Zhang et al. (2015) is the only approach that proposes equivariant learning for time-series data. They propose to learn feature representations equivariant to vocal tract length changes –an inherent symmetry of speech. However, vocal tract length changes do not conform to the mathematical definition of a group, making this equivariance only an approximate estimation. Interestingly, vocal tract length changes can be characterized by specific (scale, translation) tuples. Consequently, considering equivariance to the scale-translation group implicitly describes vocal tract length changes as well as many other symmetries encountered in audio, speech and other time-series modalities.

## 3 Background

This work assumes a basic familiarity with the concepts of a group, a subgroup and a group action. For those who may not be acquainted with these terms, we introduce these terms in Appx. A.

### 3.1 Group equivariance, group invariance and symmetry preservation

**Group equivariance.** Group equivariance is the property of a mapping to respect the transformations in a group. We say that a map is equivariant to a group if a transformation of the input by elements of the group leads to an equivalent transformation of the output (Fig. 1a). Formally, for a group $\mathcal{G}$ with elements $g \in \mathcal{G}$ acting on a set $\mathcal{X}$, and a mapping $\phi : \mathcal{X} \rightarrow \mathcal{X}$, we say that $\phi$ is equivariant to $\mathcal{G}$ if:

$$\phi(gx) = g\phi(x), \quad \forall x \in \mathcal{X}, \forall g \in \mathcal{G}. \tag{1}$$

For example, the convolution of a signal $f : \mathbb{R} \rightarrow \mathbb{R}$ and a kernel $\psi : \mathbb{R} \rightarrow \mathbb{R}$ is *equivariant to the group of translations* –or *translation equivariant*– because when the input is translated, its convolutional descriptors are equivalently translated, i.e., $(\psi * \mathcal{L}_t f) = \mathcal{L}_t(\psi * f)$, with $\mathcal{L}_t$ a translation operator by $t$: $\mathcal{L}_t f(x) = f(x-t)$.

**Group invariance.** Group invariance is a special case of group equivariance in which the output of the map is equal for all transformations of the input (Fig. 1b). Formally, for a group $\mathcal{G}$ with elements $g \in \mathcal{G}$ acting on a set $\mathcal{X}$, and a mapping $\phi : \mathcal{X} \rightarrow \mathcal{X}$, we say that $\phi$ is invariant to $\mathcal{G}$ if:

$$\phi(gx) = \phi(x), \quad \forall x \in \mathcal{X}, \forall g \in \mathcal{G}. \tag{2}$$

*Relation to symmetry preservation.* A symmetry-preserving mapping preserves the symmetries of the input. That is, if the input has certain symmetries, e.g., translation, rotation, scale, these symmetries will also be present in the output. Since symmetries are mathematically described as groups, it follows that group equivariant mappings preserve the symmetries of the group to which the mapping is equivariant. In contrast, invariant mappings *do not* preserve symmetry, as they remove all symmetric information from the input.

### 3.2 Symmetry-preserving mappings: The group and the lifting convolution

When talking about (linear) symmetry-preserving mappings, we are obliged to talk about the group convolution. Previous work has shown that group convolutions are the most general class of group equivariant linear maps (Cohen et al., 2019). Hence, it holds that any linear equivariant map is in fact a group convolution.

**Group convolution.** Let $f : \mathcal{G} \rightarrow \mathbb{R}$ and $\psi : \mathcal{G} \rightarrow \mathbb{R}$ be a scalar-valued signal and convolutional kernel defined on a group $\mathcal{G}$. The group convolution $(*_{\mathcal{G}})$ between $f$ and $\psi$ is given by:

$$(f *_{\mathcal{G}} \psi)(g) = \int_{\mathcal{G}} f(\gamma) \mathcal{L}_g \psi(\gamma) \, \mathrm{d}\mu_{\mathcal{G}}(\gamma) = \int_{\mathcal{G}} f(\gamma) \psi \left( g^{-1} \gamma \right) \, \mathrm{d}\mu_{\mathcal{G}}(\gamma). \tag{3}$$

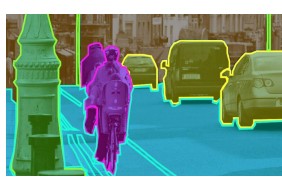 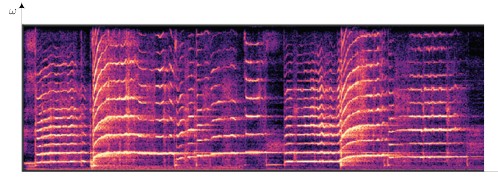

Figure 2: Locality of visual and auditory objects. Whereas visual objects are local (left), auditory objects are not. The latter often cover large parts of the frequency axis in a sparse manner (right).

where $g, \gamma \in \mathcal{G}$, $\mathcal{L}_g \psi(\gamma) = \psi\left(g^{-1}\gamma\right)$ is the action of the group $\mathcal{G}$ on the kernel $\psi$, and $\mu_{\mathcal{G}}(\gamma)$ is the (invariant) Haar measure of the group $\mathcal{G}$ for $\gamma$. Notably, the group convolution generalizes the translation equivariance of convolutions to general groups. The group convolution is equivariant in the sense that for all $\gamma, g \in \mathcal{G}$,

$$\mathcal{L}_g(f *_{\mathcal{G}} \psi)(\gamma) = (\mathcal{L}_g f *_{\mathcal{G}} \psi)(\gamma), \text{ with } \mathcal{L}_g f(\gamma) = f\left(g^{-1}\gamma\right). \tag{4}$$

**The lifting convolution.** In practice, the input signals $f$ might not be readily defined on the group of interest $\mathcal{G}$, but on a sub-domain thereof $\mathcal{X}$, i.e., $f : \mathcal{X} \to \mathbb{R}$. For example, time-series are defined on $\mathbb{R}$ although we might want to consider larger groups such as the scale-translation group. Hence, we require a symmetry-preserving mapping from $\mathcal{X}$ to $\mathcal{G}$ that *lifts* the input signal to $\mathcal{G}$ to use group convolutions. This operation is called a *lifting convolution*. Formally, with $f : \mathcal{X} \to \mathbb{R}$ and $\psi : \mathcal{X} \to \mathbb{R}$ a scalar-valued signal and convolutional kernel defined on $\mathcal{X}$, and $\mathcal{X}$ a sub-group of $\mathcal{G}$, the lifting convolution $(*_{\mathcal{G}\uparrow})$ is a mapping from functions on $\mathcal{X}$ to functions on $\mathcal{G}$ defined as:

$$(f *_{\mathcal{G}\uparrow} \psi)(g) = \int_{\mathcal{X}} f(x)\mathcal{L}_g\psi(x) \, \mathrm{d}\mu_{\mathcal{G}}(x) = \int_{\mathcal{X}} f(x)\psi(g^{-1}x) \, \mathrm{d}\mu_{\mathcal{G}}(x) \tag{5}$$

Note that, the lifting convolution is also group equivariant mapping. That is, $\mathcal{L}_g(f *_{\mathcal{G}\uparrow} \psi) = (\mathcal{L}_g f *_{\mathcal{G}\uparrow} \psi)$.

## 4 The problem of learning 2D convolutional kernels on the time-frequency plane

CNNs have been a major breakthrough in computer vision, yielding startling results in countless applications. Due to their success, several works have proposed to treat *spectro-temporal representations* –representations on the time-frequency plane– as 2D images and learn 2D CNNs on top. In this section, we delve into the differences between visual and spectro-temporal representations, and assess the suitability of training 2D CNNs directly on top of spectro-temporal representations. Our analysis suggest that treating spectro-temporal representations as images and learning 2D CNNs on top might not be adequate for effective time-series learning.

To enhance clarity, we define spectro-temporal representations in separate gray boxes throughout the section to avoid interrupting the reading flow. Those already familiar with these concepts may skip these boxes.

**Spectro-temporal representations.** Let $f(t) \in \mathrm{L}^2(\mathbb{R})$ be a square integrable function on $\mathbb{R}$. An spectro-temporal representation $\Phi[f](t, \omega) : \mathbb{R}^2 \to \mathbb{C}$ of $f$ is constructed by means of a *linear time-frequency transform* $\Phi$ that correlates the signal $f$ with a dictionary $\mathcal{D}$ of localized *time-frequency atoms* $\mathcal{D} = \{\phi_{t,\omega}\}_{t \in \mathbb{R}, \omega \in \mathbb{R}}$, $\phi_{t,\omega} : \mathbb{R} \to \mathbb{C}$ of finite energy and unitary norm, i.e., $\phi_{t,\omega} \in \mathrm{L}^2(\mathbb{R})$, $\|\phi_{t,\omega}\|^2 = 1$, $\forall t \in \mathbb{R}, \omega \in \mathbb{R}$. The resulting spectro-temporal representation $\Phi[f]$ is given by:

$$\Phi[f](t, \omega) = \langle f, \phi_{t,\omega} \rangle = \int_{\mathbb{R}} f(\tau)\phi_{t,\omega}^*(\tau) \, \mathrm{d}\tau, \tag{6}$$

with $\phi^*$ the complex conjugate of $\phi$, and $\langle \cdot, \cdot \rangle$ the dot product of its arguments. Using different time-frequency components $\phi_{t,\omega}$, spectro-temporal representations with different properties can be obtained.

### 4.1 Fundamental differences between visual representations and spectro-temporal representations

There exist two fundamental distinctions between visual data and spectro-temporal representations, which are universal to all spectro-temporal representations: (*i*) locality and (*ii*) transparency. Unlike visual data, auditory signals exhibit strong *non-local* characteristics. Auditory signals consist of auditory objects, e.g., spoken words, which contain components resonating at multiple non-local frequencies known as the *harmonics of the signal*. Consequently, the spectro-temporal representations of auditory objects often occupy a significant portion of the time-frequency plane –particularly along the frequency axis $(\omega)$– in a sparse manner

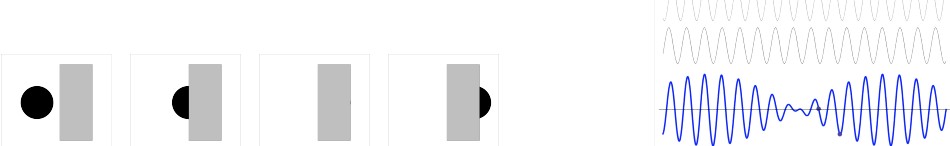

Figure 3: Occlusion and superposition. Visual objects occlude each other when they appear simultaneously at a given position (left). Auditory objects, instead, superpose at all shared positions (right).

(Fig. 2). Furthermore, when considering auditory signals comprising multiple auditory objects, these objects exhibit a phenomenon known as *superposition*. This property is notably different from visual data, where visual objects in the same location *occlude* one another, resulting in only the object closest to the camera being visible (Fig. 3). This inherent property of sound is colloquially referred to as *transparency*.

### 4.2 The problem of learning 2D kernels on short-time Fourier spectro-temporal representations

The short-time Fourier transform constructs a representation in which a signal is decomposed in terms of its correlation with time-frequency atoms of constant time and frequency resolution. As a result, it is effective as long as the signal $f$ does not exhibit *transient behavior* –components that evolve quickly over time– with some waveform structures being very localized in time and others very localized in frequency.

**The short-time Fourier transform.** The short-time Fourier transform $\mathcal{S}$ –also called *windowed Fourier transform*– is a linear time-frequency transform that uses a dictionary of time-frequency atoms $\phi_{t,\omega}(\tau)=w(\tau-t)e^{-i\omega\tau}$, $t\in\mathbb{R}$, $\omega\in\mathbb{R}$, constructed with a symmetric window $w(\tau)$ of local support shifted by $t$ and modulated by the frequency $\omega$. The spectro-temporal representation $\mathcal{S}[f]$ is given by:

$$\mathcal{S}[f](t,\omega) = \langle f, \phi_{t,\omega} \rangle = \int_{\mathbb{R}} f(\tau)\phi_{t,\omega}^*(\tau)\,d\tau = \int_{\mathbb{R}} f(\tau)w(\tau-t)e^{-i\omega\tau}\,d\tau. \tag{7}$$

Intuitively, the short-time Fourier transform divides the time-frequency plane in tiles of equal resolution, whose value is given by the correlation between $f$ and the time-frequency atom $\phi_{t,\omega}$ (Fig. 4a).

Nevertheless, decades of research in psychology and neuroscience have shown that humans largely rely in the transient behavior of auditory signals to distinguish auditory objects (Cherry, 1953; van Noorden et al., 1975; Moore & Gockel, 2012). In addition, it has been shown that the human auditory system has high spectral resolution at low-frequencies and high temporal resolution at higher frequencies (Stevens et al., 1937; Santoro et al., 2014; Bidelman & Khaja, 2014). For example, a semitone at the bottom of the piano scale ($\sim$30Hz) is of about 1.5Hz, while at the top of the musical scale ($\sim$5kHz) it is of about 200Hz. These properties of the human auditory signal largely contrast both with ($i$) the inability of the short-time Fourier transform to detect transient signals, as well as with ($ii$) its constant spectro-temporal resolution.

To account for these differences, improved spectro-temporal representations on top of the short-time Fourier transform have been proposed such as log-Mel spectrograms (Stevens et al., 1937; Furui, 1986). These developments revolve around transforming the frequency axis of the short-time Fourier transform in a logarithmic scale, thereby compressing the frequency axis and better aligning with the spectro-temporal resolution of

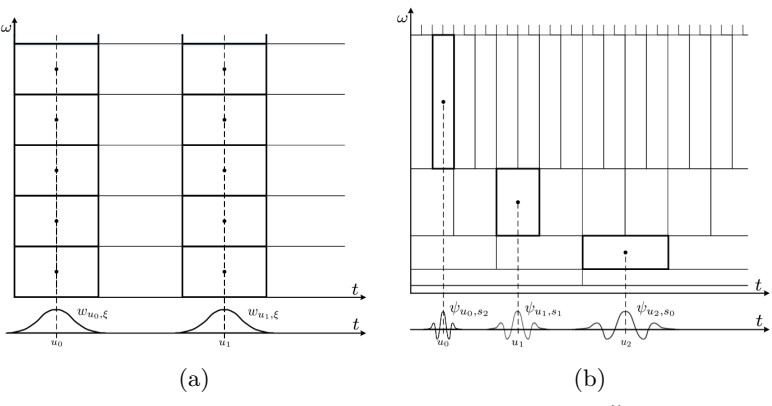

(a)             (b)

Figure 4: Tiling of the time-frequency plane for the short-time Fourier transform (Fig. 4a) and the wavelet transform (Fig. 4b). The short-time Fourier transform divides the time-frequency plane in tiles of equal resolution. This makes it adequate for signals without transient behaviour. The Wavelet transform, on the other hand, divides the time-frequency plane on tiless of changing spectro-temporal resolution. This allows it to represent detect highly localized events both on time and frequency.

the human auditory system. Consequently, this adjustment enables local structures, e.g., 2D convolutional kernels, to better capture non-local relationships (Ullrich et al., 2014; Choi et al., 2016; Xu et al., 2018). However, despite their improved learning characteristics, these spectro-temporal representations remain *incomplete* due to their inability to modify the constant temporal resolution of the short-time Fourier transform.

### 4.3 The problem of learning 2D kernels on Wavelet spectro-temporal representations

In contrast to the short-time Fourier transform, the Wavelet transform constructs a spectro-temporal representation in terms of correlations with time-frequency atoms, *whose time and frequency resolution change*. As a result, the resulting decomposition of the time-frequency plane allows the wavelet transform to *correctly describe signals with transient behaviour with localized components both on time and frequency* (Fig. 4b).

> **The wavelet transform.** The wavelet transform $\mathcal{W}$ is a linear time-frequency transform that uses a dictionary of time-frequency atoms $\phi_{t,\omega}(\tau) = \frac{1}{\sqrt{\omega}} \psi\left(\frac{\tau - t}{\omega}\right)$, $t \in \mathbb{R}$, $\omega \in \mathbb{R}_{\geq 0}$. The function $\psi_{t,\omega}$ is called a *Wavelet* and satisfies the properties of having zero mean, i.e., $\int \psi_{t,\omega}(\tau) \mathrm{d}\tau = 0$,s and being unitary, i.e., $\|\psi_{t,\omega}\|^2 = 1$, for any $t \in \mathbb{R}$, $\omega \in \mathbb{R}_{\geq 0}$. The resulting spectro-temporal representation $\mathcal{W}[f]$ is given by:
>
> $$\mathcal{W}[f](t, \omega) = \langle f, \phi_{t,\omega} \rangle = \int_{\mathbb{R}} f(\tau) \phi_{t,\omega}^*(\tau) \, \mathrm{d}\tau = \int_{\mathbb{R}} f(\tau) \frac{1}{\sqrt{\omega}} \psi\left(\frac{\tau - t}{\omega}\right) \, \mathrm{d}\tau. \tag{8}$$
>
> Intuitively, the Wavelet transform divides the time-frequency plane in tiles of different resolutions, with high frequency resolution and low spatial resolution at low frequencies, and low frequency resolution and high spatial resolution for high frequencies (Fig. 4b).[a]
>
> ---
> [a]Importantly, it is not possible to have high frequency and spatial resolution at the same time due to the *uncertainty principle* (Gabor, 1946). It states that the joint time-frequency resolution of spectro-temporal representations is limited by a minimum surface $\sigma_{\phi,t}\sigma_{\phi,\omega} \geq \frac{1}{2}$, with $\sigma_{\phi,t}$, $\sigma_{\phi,\omega}$ the spread of the time-frequency atom $\phi$ on time and frequency.

Interestingly, the modus operandi of the wavelet transform *perfectly aligns* with the spectro-temporal resolution used by the human auditory system for the processing of auditory signals. Nevertheless, despite this resemblance, training local 2D structures, e.g., convolutional kernels, directly on the wavelet transform's output stil falls short in addressing the non-local, transparent characteristics inherent in auditory signals. Consequently, researchers have devised several strategies to overcome these challenges, e.g., by defining separable kernels that span large memory horizons along the frequency and time axis independently (Pons & Serra, 2019) or by prioritizing learning along the harmonics of a given frequency (Zhang et al., 2020).

As shown in the next section, a better alternative arises from considering the symmetries appearing in time-series data. Starting from this perspective, we are led to scale-translation equivariant mappings and find striking relationships between these family of mappings and the wavelet transform. Nevertheless, our analysis indicates that *all layers within a neural network should be symmetry preserving* –a condition not met by the methods depicted in this section. By doing so, we devise neural architectures, whose convolutional layers process the output of previous layers in a manner akin to the wavelet transform. As a result, each convolutional layer performs spectro-temporal decompositions of the input in terms of localized time-frequency atoms able to process global and localized patterns both on time and frequency.

## 5 Wavelet networks: Scale-translation equivariant learning from raw waveforms

We are interested in mappings that preserve the scale and translation symmetries of time-series. In this section, we start by tailoring lifting and group convolutions to the scale-translation group. Next, we outline the general form of Wavelet Networks and make concrete practical considerations for their implementation. At the end of this section, we formalize the relationship between Wavelet Networks and the wavelet transform, and provide a thorough analysis on the equivariance properties of common spectro-temporal transforms.

### 5.1 Scale-translation preserving mappings: group convolutions on the scale-translation group

We are interested in mappings that preserve scale and translation. By imposing equivariance to the *scale-translation* group, we guarantee that if input patterns are scaled, translated, or both, their feature representations will transform accordingly, but not be modified.

**The scale-translation group.** From a mathematical perspective scale and translational symmetries are described by the affine *scale-translation group* $\mathcal{G} = \mathbb{R} \rtimes \mathbb{R}_{\geq 0}$, which emerges from the semi-direct product of

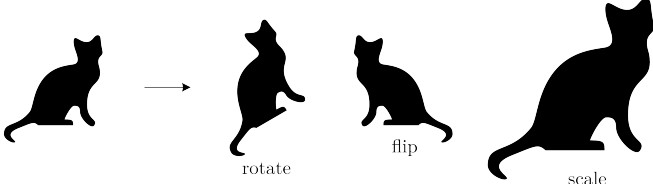

Figure 5: The action of unimodular and non-unimodular groups. Most unimodular groups, e.g., rotation, mirroring, keep the volume of the objects they act upon intact. In contrast, non-unimodular groups, e.g., scaling, change it through their action.

the translation group $\mathcal{T}=(\mathbb{R},+)$ and the scale group $\mathcal{S}=(\mathbb{R}_{\geq 0},\times)$ acting on $\mathbb{R}$. As a result, we have that the resulting group product is given by $g\cdot\gamma=(t,s)\cdot(\tau,\varsigma)=(t+s\tau,s\cdot\varsigma)$, with $t,\tau\in\mathbb{R}$ and $s,\varsigma\in\mathbb{R}_{\geq 0}$. In addition, by solving $g^{-1}\cdot g=e$, we obtain that the inverse of a group element $g=(t,s)$ is given by $g^{-1}=s^{-1}(-t,1)$.

> **Semi-direct product and affine groups.** When treating data defined on $\mathbb{R}^d$, one is mainly interested in the analysis of groups of the form $\mathcal{G}=\mathbb{R}^d\rtimes\mathcal{H}$ resulting from the *semi-direct product* ($\rtimes$) between the translation group $(\mathbb{R}^d,+)$ and an arbitrary (Lie) group $\mathcal{H}$ acting on $\mathbb{R}^d$, e.g., rotation, scale, etc. This kind of groups are called *affine groups* and their group product is defined as:
>
> $$g_1\cdot g_2=(x_1,h_1)\cdot(x_2,h_2)=(x_1+\mathcal{A}_{h_1}(x_2),h_1\cdot h_2),\tag{9}$$
>
> with $g_1=(x_1,h_1)$, $g_2=(x_2,h_2)\in\mathcal{G}$, $x_1,x_2\in\mathbb{R}^d$ and $h_1,h_2\in\mathcal{H}$. $\mathcal{A}$ denotes the action of $\mathcal{H}$ on $\mathbb{R}^d$.

**Unimodular and non-unimodular groups.** Unimodular groups, such as rotation, translation and mirroring, are groups whose action keeps the volume of the objects on which they act intact (Fig. 5). Recall that a group convolution performs an integral over the whole group (Eq. 3). Hence, for its result to be invariant over different group actions, it is required for the Haar measure to be equal for all elements of the group –therefore the name *invariant* Haar measure. Since the action of (most) unimodular groups does not alter the size of the objects on which they act, their action on infinitesimal objects keeps their size unchanged. As a consequence, for (most) unimodular groups, the Haar measure is equal to the Lebesgue measure, i.e., $\mathrm{d}\mu_{\mathcal{G}}(\gamma)=\mathrm{d}\gamma$, $\forall\gamma\in\mathcal{G}$, and therefore, it is often omitted in literature, e.g., in Cohen & Welling (2016).

In contrast, non-unimodular groups, such as the scale group and the scale-translation group, do modify the size of objects on which they act (Fig. 5 right). Consequently, their action on infinitesimal objects changes their size. As a result, the Haar measure must be treated carefully in order to obtain equivariance to non-unimodular groups (Bekkers, 2020). The Haar measure guarantees that $\mathrm{d}\mu_{\mathcal{G}}(\gamma)=\mathrm{d}\mu_{\mathcal{G}}(g\gamma)$, $\forall\,g,\gamma\in\mathcal{G}$. For the scale-translation group, it is obtained as:

$$\mathrm{d}\mu_{\mathcal{G}}(\gamma)=\mathrm{d}\mu_{\mathcal{G}}(g\gamma)=\mathrm{d}\mu_{\mathcal{G}}(t+s\tau,s\varsigma)=\mathrm{d}\mu_{\mathcal{G}}(t+s\tau)\mathrm{d}\mu_{\mathcal{G}}(s\varsigma)=\frac{1}{|s|}\mathrm{d}\tau\frac{1}{|s|}\mathrm{d}\varsigma,\tag{10}$$

where $g=(t,s),\gamma=(\tau,\varsigma)\in\mathcal{G}$, $t,\tau\in\mathbb{R}$, $s,\varsigma\in\mathbb{R}_{>0}$; $\mathrm{d}\tau$, $\mathrm{d}\varsigma$ are the Lebesgue measure of the respective spaces; and $|s|$ depicts the determinant of the matrix representation of the group element.[1] Intuitively, the Haar measure counteracts the growth of infinitesimal elements resulting from the action of $s$ on $\mathbb{R}\times\mathbb{R}_{>0}$.

**Scale-translation group convolutions.** The general formulation of the group convolution is given in Eq. 3. Interestingly, the scale-translation group has additional properties with which this formulation can be simplified. In particular, by taking advantage of the fact that the scale-translation group is an *affine* group $\mathcal{G}=\mathbb{R}\rtimes\mathcal{S}$, with $\mathcal{S}=(\mathbb{R}_{>0},\times)$, as well as of the definition of the Haar measure for the scale-translation group in Eq. 10 we can reformulate the group convolution for the scale-translation group as:

$$(f*_{\mathcal{G}}\psi)(g)=\int_{\mathcal{G}}f(\gamma)\psi(g^{-1}\gamma)\,\mathrm{d}\mu_{\mathcal{G}}(\gamma)$$

$$(f*_{\mathcal{G}}\psi)(t,s)=\int_{\mathcal{S}}\int_{\mathbb{R}}f(\tau,\varsigma)\psi\left((t,s)^{-1}(\tau,\varsigma)\right)\frac{1}{|s|}\mathrm{d}\tau\frac{1}{|s|}\mathrm{d}\varsigma=\int_{\mathcal{S}}\int_{\mathbb{R}}f(\tau,\varsigma)\frac{1}{s^2}\psi\left(s^{-1}(\tau-t,\varsigma)\right)\,\mathrm{d}\tau\,\mathrm{d}\varsigma$$

$$=\int_{\mathcal{S}}\int_{\mathbb{R}}f(\tau,\varsigma)\frac{1}{s^2}\mathcal{L}_s\psi(\tau-t,\varsigma)\,\mathrm{d}\tau\,\mathrm{d}\varsigma=\int_{\mathcal{S}}\left(f*_{\mathbb{R}}\frac{1}{s^2}\mathcal{L}_s\psi\right)(t,\varsigma)\,\mathrm{d}\varsigma\tag{11}$$

---

[1]A member $s$ of the scale group $\mathbb{R}_{>0}$ acting on a $N$-dimensional space is represented a matrix $\mathrm{diag}(s,...,s)$. Since, its determinant $s^N$ depends on the value of the group element $s$, the factor $\frac{1}{|s|}=\frac{1}{s^N}$ in Eq. 10 cannot be omitted.

Scale–Translation Lifting Convolution:

Scale–Translation Group Convolution:

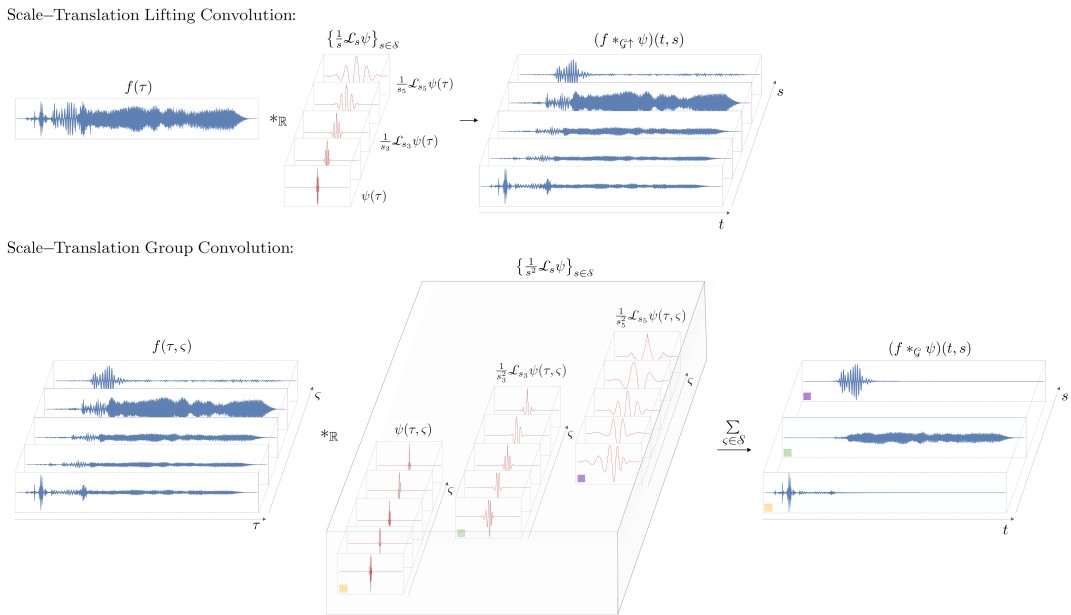

Figure 6: Scale-translation lifting and group convolution. The lifting convolution can be seen a set of 1D convolutions with a bank of scaled convolutional kernels $\frac{1}{s}\mathcal{L}_s\psi$, and the group convolution can be seen as a set of 1D convolutions with a bank of scaled convolutional kernels $\frac{1}{s^2}\mathcal{L}_s\psi$, followed by an integral over scales $\varsigma \in \mathbb{R}$. Their main difference is that, for group convolutions, the input $f$ and the convolutional kernel $\psi$ are functions on the scale-translation group whereas for lifting convolutions these are functions on $\mathbb{R}$. Lifting and group convolutions can be seen as spectro-temporal decompositions with large values of $s$ relating to coarse features and small values to finer features.

where $g=(t,h)$, $\gamma=(\tau,\varsigma) \in \mathcal{G}$, $t,\tau \in \mathbb{R}$, and $s,\varsigma \in \mathbb{R}_{>0}$; and $\mathcal{L}_s\psi(\tau,\varsigma)=\psi\left(s^{-1}(\tau,\varsigma)\right)$ is the (left) action of the scale group $\mathcal{S}$ on a convolutional kernel $\psi : \mathbb{R} \times \mathbb{R}_{>0} \to \mathbb{R}$ defined on the scale-translation group. In other words, for the scale-translation group, *the group convolution can be seen as a set of 1D convolutions with a bank of scaled convolutional kernels $\{\frac{1}{s^2}\mathcal{L}_s\psi\}_{s\in\mathcal{S}}$, followed by an integral over scales $\varsigma \in \mathbb{R}$* (Fig. 6, bottom).

**Scale-translation lifting convolution.** Like the group convolution, the lifting convolution can also be simplified by considering the properties of the scale-translation group. In particular, we can rewrite it as:

$$(f *_{\mathcal{G}\uparrow} \psi)(g) = \int_{\mathcal{X}} f(x)\psi(g^{-1}x) \, \mathrm{d}\mu_{\mathcal{G}}(x) = \int_{\mathbb{R}} f(\tau)\psi(g^{-1}\tau)\mathrm{d}_{\mathcal{G}}(\tau)$$

$$(f *_{\mathcal{G}\uparrow} \psi)(t,s) = \int_{\mathbb{R}} f(\tau)\psi((t,s)^{-1}\tau) \, \frac{1}{|s|}\mathrm{d}\tau = \int_{\mathbb{R}} f(\tau) \, \frac{1}{s}\psi\left(s^{-1}(\tau - t)\right) \, \mathrm{d}\tau = \left(f *_{\mathbb{R}} \frac{1}{s}\mathcal{L}_s\psi\right)(t) \quad (12)$$

where $g=(t,h)$, $\gamma=(\tau,\varsigma) \in \mathcal{G}$, $t,\tau \in \mathbb{R}$, and $s,\varsigma \in \mathbb{R}_{>0}$; and $\mathcal{L}_s\psi(\tau,\varsigma)=\psi\left(s^{-1}(\tau,\varsigma)\right)$ is the (left) action of the scale group $\mathcal{S}$ on a 1D convolutional kernel $\psi : \mathbb{R} \to \mathbb{R}$. In other words, for the scale-translation group, *the lifting convolution can be seen as a set of 1D convolutions with a bank of scaled convolutional kernels $\{\frac{1}{s}\mathcal{L}_s\psi\}_{s\in\mathcal{S}}$* (Fig. 6, top). Note that the Haar measure imposes a normalization factor of $\frac{1}{s^2}$ for group convolutions and of $\frac{1}{s}$ for the lifting convolution. This is because space on which the group convolution is performed ($\mathbb{R} \rtimes \mathbb{R}_{>0}$) has an additional dimension relative to the space on which the lifting convolution is performed ($\mathbb{R}$).

## 5.2 Wavelet Networks: architecture and practical implementation

The general architecture of our proposed Wavelet networks is shown in Fig. 7. Wavelet networks consist of several stacked layers that respect scale and translation. They consist of a lifting group convolution layer that lifts input time-series to the scale-translation group, followed by arbitrarily many group convolutional layers. At the end of the network, a global pooling layer is used to produce scale-translation invariant representations. Due to their construction, Wavelet networks make sure that common neural operations, e.g., point-wise nonlinearities, do not disrupt scale and translation equivariance. This in turn, makes them broadly applicable and easily extendable to other existing neural archi-

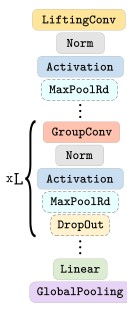

Figure 7: Wavelet networks.

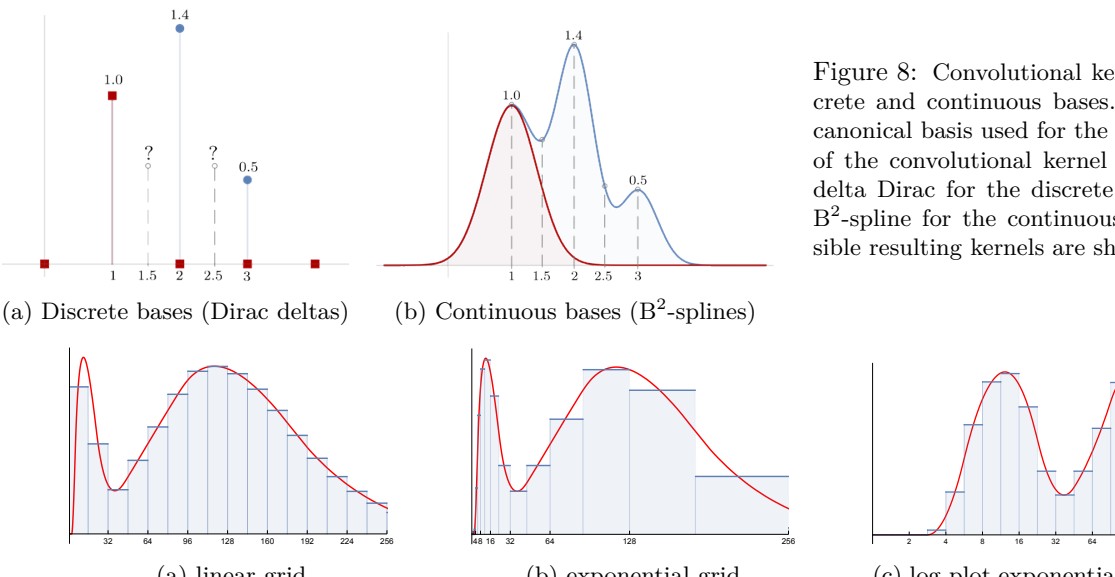

Figure 8: Convolutional kernels on discrete and continuous bases. In red the canonical basis used for the construction of the convolutional kernel is shown: a delta Dirac for the discrete case, and a $B^2$-spline for the continuous case. Possible resulting kernels are shown in blue.

(a) Discrete bases (Dirac deltas)  (b) Continuous bases ($B^2$-splines)

(a) linear grid  (b) exponential grid  (c) log-plot exponential grid

Figure 9: Riemann integration of functions on $\mathbb{R}_{>0}$ using linear (9a) and exponential grids (9b, 9c).

tectures, e.g., ResNets (He et al., 2016), U-Nets (Ronneberger et al., 2015).

### 5.2.1 Group convolutional kernels on continuous bases

Although our previous derivations build upon continuous functions, in practice, computations are performed on discretized versions of these functions. Continuous bases have proven advantageous for the construction of group convolutions as the action of relevant groups often impose transformations not well-defined for discrete bases (Weiler et al., 2018; Bekkers et al., 2018; Weiler & Cesa, 2019). For instance, in the context of scale-translations, scaling a kernel $[w_1, w_2, w_3]$ by a factor of two results in a filter $[w_1, w_{1.5}, w_2, w_{2.5}, w_3]$ wherein the introduced values $[w_{1.5}, w_{2.5}]$ do not exist in the original basis (Fig. 8a).

The most adopted solution to address this problem is interpolation, i.e., deriving the value of $[w_{1.5}, w_{2.5}]$ based on the neighbouring known pixels. However, interpolation introduces spurious artifacts which are particularly severe for small kernels. Instead, we adopt an alternative approach: we define convolutional kernels directly on a continuous basis (Fig. 8b). Drawing from the resemblance of gammatone filters – strongly motivated by the physiology of the human auditory system for the processing and recognition of auditory signals (Johannesma, 1972; Hewitt & Meddis, 1994; Lindeberg & Friberg, 2015a)– to $B^2$-splines, we parameterize our filters within a $B^2$-spline basis as in Bekkers (2020). As a result, our convolutional filters are parameterized as a linear combination of shifted $B^2$-splines $\psi(\tau) \coloneqq \sum_{i=1}^N w_i B^2(\tau - \tau_i)$, rather than the commonly used shifted Dirac delta's basis $\psi(\tau) \coloneqq \sum_{i=1}^N w_i \delta(\tau - \tau_i)$.

### 5.2.2 Constructing a discrete scale grid

From the response of the lifting layers onward, the feature representations of wavelet networks possess an additional axis $s \in \mathbb{R}_{>0}$. Just like the spatial axis, this axis must be discretized in order to perform computational operations. That is, we must approximate the scale axis $\mathbb{R}_{>0}$ by a finite set of discrete scales $\{s\}_{s=s_{\min}}^{s_{\max}}$. Inspired by previous work, we approximate the scale axis with a *dyadic set* $\{2^j\}_{j=j_{\min}}^{j_{\max}}$ (Mallat, 1999; Lindeberg & Friberg, 2015b; Worrall & Welling, 2019). Dyadic sets resemble the spectro-temporal resolution of the human auditory system, and are widely used for discrete versions of the wavelet transform.

**Integrating on exponential grids.** A subtlety arises with respect to integrating over the scale axis when implementing the continuous theory in a discrete setting that is suitable for numerical computations. The group convolutions include scale correction factors as part of the Haar measure, which makes the integration invariant to actions along the scale axis. That is, the integral of a signal $f(s)$ over scale is the same as that of the same signal $f(z^{-1}s)$, whose scale is changed by a factor $z \in \mathbb{R}_{>0}$:

$$\int_{\mathbb{R}_{>0}} f(z^{-1}s)\tfrac{1}{s}\mathrm{d}s \overset{s \to zs}{=} \int_{\mathbb{R}_{>0}} f(z^{-1}s)\tfrac{1}{zs}\mathrm{d}zs = \int_{\mathbb{R}_{>0}} f(s)\tfrac{1}{s}\mathrm{d}s. \tag{13}$$

We can translate the scale integration to the discrete setting via Riemann integrals, where we sample the function on a grid and take the weighted sum of these values with weights given by the bin-width:

$$\int_{\mathbb{R}_{>0}} f(s)\frac{1}{s}\mathrm{d}s \approx \sum_i f(s_i)\frac{1}{s_i}\Delta_i. \tag{14}$$

When the scale grid is linear, the bin-widths $\Delta_i$ are constant, as depicted in Fig. 9a. When the scale grid is exponential, e.g., $s_i = b^{i-1}$ with $b$ some base factor, the bin widths are proportional to the scale values at the grid points, i.e., $\Delta_i \propto s_i$ (Fig. 9b). In this setting, the factor $\frac{1}{s_i}$ cancels out (up to some constant) with the bin width $\Delta_i$, and integration is simply done by summing the values sampled on the scale grid. Consequently, when working with an exponential grid along the scale axis, the factor in the group convolutions (Eq. 11) becomes $\frac{1}{s}$ instead of $\frac{1}{s^2}$. It is worth mentioning that using an exponential grid is the natural thing to do when dealing with the scale group. The scale group is a multiplicative group with a natural distance between group elements $z, s \in \mathbb{R}_{>0}$ defined by $\| \log z^{-1}s \|$. Consequently, on an exponential grid, the grid points are spaced uniformly with respect to this distance, as illustrated in Fig. 9c.

**Defining the discrete scale grid.** In practice, Wavelet networks must define the number of scales $N_s$ to be considered in the dyadic set as well as its limits $s_{\min}, s_{\max}$. Fortunately, it turns out that these values are related to the spatial dimension of the input $f$ itself, and thus, we can use it to determine these values.

Let us consider a signal $f$ and a convolutional kernel $\psi$ sampled on discrete grids $[1, N_f], [1, N_\psi] \subset \mathbb{Z}$ of sizes $N_f$, and $N_\psi$, respectively. When we re-scale the convolutional kernel $\psi$, we are restricted $(i)$ at the bottom of the scale axis by the Nyquist criterion, and $(ii)$ at the top of the scale by the scale for which the filter becomes constant in an interval of $N_f$ samples. The Nyquist criterion is required to avoid aliasing and intuitively restricts us to a compression factor on $\psi$ such that it becomes as big as 2 grid samples. On the other hand, by having $\psi$ re-scaled to an extreme to which it is constant in the support of the input signal $f$, the kernel will only be able to perform average operations.

*Considerations regarding computational complexity.* Note that the computational cost of Wavelet networks increases linearly with the number of scales considered. Hence, it is desirable to reduce the number of scales used as much as possible. To this end, we reason that using scales for which the sampled support of $\psi$ is smaller than $N_\psi$ is unnecessary as the functions that can be described at those scales can also be described –and learned– at the unscaled resolution of the kernel $s=1$. Therefore, we define the minimum scale as $s_{\min}=1$. Furthermore, we reason that using scales for which the support of the filter overpasses that of the input, i.e., $N_f \leq N_\psi$, is also suboptimal, as the values outside of the region $[1, N_f]$ are unknown. Therefore, we consider the set of sensible scales to be given by the interval $[1, \frac{N_f}{N_\psi}]$. In terms of a dyadic set $\{2^j\}_{j=j_{\min}}^{j_{\max}}$, this corresponds to the $j$-values given by the interval $[0, 1, 2, ..., j_{\max}$ s.t. $N_\psi\ 2^{j_{\max}} \leq N_f]$.

*Effect of downsampling on the scale grids used.* Neural architectures utilize pooling operations, e.g., max-pooling, to reduce the spatial dimension of the input as a function of depth. Following the rationale outlined in the previous paragraph, we take advantage of these reductions to reduce the number of scales that representations at a given depth should use. Specifically, we use the factor of downsampling as a proxy for the number of scales that can be disregarded. For example, if we use a pooling of 8 at a given layer, subsequent layers should reduce the number of scales considered by the same factor, i.e., $2^3$. For a set of dyadic scales before a pooling layer given by $\{2^j\}_{j=j_{\min}}^{j_{\max}}$ and a pooling layer of factor $2^p$, the set of dyadic scales considered after pooling will be given by $\{2^j\}_{j=j_{\min}}^{j_{\max}-p}$.

### 5.2.3 Imposing wavelet structure to the learned convolutional kernels

In classical spectro-temporal analysis, wavelets are designed to have unit norm $\|\psi\|^2=1$ and zero mean $\int \psi(\tau)\, \mathrm{d}\tau=0$. These constraints are useful for both theoretical and practical reasons including energy preservation, numerical stability and the ability to act as band-pass filters (Mallat, 1999). Since Wavelet networks construct time-frequency representations of the input, we experiment with an additional regularization loss that encourages the learned convolutional kernels to behave like wavelets. First, we note that lifting and group convolutions inherently incorporate a normalization term $-\frac{1}{s}, \frac{1}{s^2}-$ in their definitions. Therefore, the normalization criterion is inherently satisfied. To encourage the learned kernels to have zero mean, we formulate a regularization term that promote this behaviour. Denoting $\psi_d$ as the convolutional kernel at the

$d$-th layer of a neural network with D convolutional layers, the regularization term $\mathcal{L}_{\text{wavelet}}$ is defined as:

$$\mathcal{L}_{\text{wavelet}} = \sum_{d=1}^{D} \|\text{mean}(\psi_d)\|^2. \tag{15}$$

Interestingly, we observe that enforcing wavelet structure in the learned convolutional kernels consistently yields improved performance across all tasks considered (Sec. 6). This result underscores the potential value of integrating insights from classical signal processing, e.g., spectro-temporal analysis (Scharf, 1991; Mallat, 1999; Daubechies, 2006), in the design of deep learning architectures.

### 5.3 Wavelet networks perform nested non-linear time-frequency transforms

Interestingly, we can use spectro-temporal analysis to understand the modus operandi of wavelet networks. Our analysis reveals that wavelet networks perform nested time-frequency transforms interleaved with point-wise nonlinearities. In this process, each time-frequency transform emerges as a linear combination of parallel wavelet-like transformations of the input computed with learnable convolutional kernels $\psi$.

**The relation between scale-translation equivariant mappings and the wavelet transform.** The wavelet transform shows many similarities to the scale-translation group and lifting convolutions (Grossmann et al., 1985). In fact, by analyzing the definition of the wavelet transform (Eq. 8), we obtain that the Wavelet transform is equivalent to a lifting group convolution (Eq. 12 with $\omega{=}s$) up to a normalization factor $\frac{1}{\sqrt{\omega}}$:

$$\mathcal{W}[f](t,\omega) = \int_{\mathbb{R}} f(\tau) \frac{1}{\sqrt{\omega}} \psi\left(\frac{\tau-t}{\omega}\right) \, d\tau = \int_{\mathbb{R}} f(\tau) \frac{1}{\sqrt{\omega}} \psi\left(\omega^{-1}(\tau-t)\right) \, d\tau$$

$$= \int_{\mathbb{R}} f(\tau) \frac{1}{\sqrt{\omega}} \mathcal{L}_\omega \psi\left(\tau-t\right) \, d\tau = \left(f *_{\mathbb{R}} \frac{1}{\sqrt{\omega}} \mathcal{L}_\omega \psi\right)(t) = \frac{1}{\sqrt{\omega}} \left(f *_{\mathcal{G}\uparrow} \psi\right)(t,\omega). \tag{16}$$

Furthermore, if we let the input $f$ be a function defined on the scale-translation group, and let $\omega$ act on this group according to the group structure of the scale-translation group, we have that the scale-translation group convolution is equivalent to a Wavelet transform whose input has been obtained by a previously applied Wavelet transform, up to a normalization factor $\frac{1}{\omega\sqrt{\omega}}$:

$$\mathcal{W}[f](t,\omega) = \int_{\mathbb{R}_{>0}} \int_{\mathbb{R}} f(\tau,\varsigma) \frac{1}{\sqrt{\omega}} \psi\left(\omega^{-1}(\tau-t),\varsigma\right) \, d\tau \, d\varsigma = \int_{\mathbb{R}_{>0}} \int_{\mathbb{R}} f(\tau,\varsigma) \frac{1}{\sqrt{\omega}} \mathcal{L}_\omega \psi\left(\tau-t,\varsigma\right) \, d\tau \, d\varsigma$$

$$= \int_{\mathbb{R}_{>0}} \left(f *_{\mathbb{R}} \frac{1}{\sqrt{\omega}} \mathcal{L}_\omega \psi\right)(t,\varsigma) \, d\varsigma = \frac{1}{\omega\sqrt{\omega}} \left(f *_{\mathcal{G}} \psi\right)(t,\omega) \tag{17}$$

In other words, lifting and group convolutions on the scale-translation group can be interpreted as linear time-frequency transforms that adopt time-frequency plane tiling akin wavelet transform (Fig. 4b), for which the group convolution accepts wavelet-like spectro-temporal representations as input.

> **Equivariance properties of common time-frequency transforms.** For completeness, we also analyze the equivariance properties of common time-frequency transforms and their normalized representations, e.g., spectrogram. Careful interpretations and proofs are provided in Appx. B.
>
> Let $\mathcal{L}_{t_0} f = f(t-t_0)$ and $\mathcal{L}_{s_0} f(t) = f(s_0^{-1}t)$, $t_0 \in \mathbb{R}$, $s_0 \in \mathbb{R}_{>0}$, be translation and scaling operators. The Fourier, short-time Fourier and Wavelet transform of $\mathcal{L}_{t_0} f$ and $\mathcal{L}_{s_0} f$, $f \in \mathrm{L}^2(\mathbb{R})$, are given by:
>
> - **Fourier Transform:**
>
> $$\mathcal{F}[\mathcal{L}_{t_0} f](\omega) = e^{-i\omega t_0} \mathcal{F}[f](\omega) \qquad \rightarrow |\mathcal{F}[\mathcal{L}_{t_0} f](\omega)|^2 = |\mathcal{F}[f](\omega)|^2 \tag{18}$$
>
> $$\mathcal{F}[\mathcal{L}_{s_0} f](\omega) = s_0 \mathcal{L}_{s_0^{-1}} \mathcal{F}[f](\omega) \qquad \rightarrow |\mathcal{F}[\mathcal{L}_{s_0} f](\omega)|^2 = |s_0|^2 |\mathcal{L}_{s_0^{-1}} \mathcal{F}[f](\omega)|^2 \tag{19}$$
>
> - **Short-Time Fourier Transform:**
>
> $$\mathcal{S}[\mathcal{L}_{t_0} f](t,\omega) = e^{-i\omega t_0} \mathcal{L}_{t_0} \mathcal{S}[f](t,\omega) \qquad \rightarrow |\mathcal{S}[\mathcal{L}_{t_0} f](t,\omega)|^2 = |\mathcal{L}_{t_0} \mathcal{S}[f](t,\omega)|^2 \tag{20}$$
>
> $$\mathcal{S}[\mathcal{L}_{s_0} f](t,\omega) \approx s_0 \, \mathcal{S}[f](s_0^{-1}t, s_0\omega) \qquad \rightarrow |\mathcal{S}[\mathcal{L}_{s_0} f](t,\omega)|^2 \approx |s_0|^2 |\mathcal{S}[f](s_0^{-1}t, s_0\omega)|^2 \; ^{(*)} \tag{21}$$

- **Wavelet Transform:**

$$\mathcal{W}[\mathcal{L}_{t_0}[f]](t,\omega) = \mathcal{L}_{t_0}\mathcal{W}[f](t,\omega) \qquad \rightarrow |\mathcal{W}[\mathcal{L}_{t_0}f](t,\omega)|^2 = |\mathcal{L}_{t_0}\mathcal{W}[f](t,\omega)|^2 \qquad (22)$$

$$\mathcal{W}[\mathcal{L}_{s_0}f](t,\omega) = \sqrt{s_0}\,\mathcal{L}_{s_0}\mathcal{W}[f](t,\omega) \qquad \rightarrow |\mathcal{W}[\mathcal{L}_{s_0}f](t,\omega)|^2 = |\mathcal{L}_{s_0}\mathcal{W}[f](t,\omega)|^2 \qquad (23)$$

(*) Eq. 21 only approximately holds for large windows (see Appx. B.2 for a detailed explanation).

In other words, the Wavelet transform and the scalogram $|\mathcal{W}[\cdot]|^2$ are the only time-frequency representations that exhibit both translation and scaling equivariance in a practical way.

**Wavelet networks apply parallel time-frequency transforms with learned bases at every layer.** So far, our analysis has been defined for scalar-valued input and convolutional kernels. However, in practice, convolutional layers perform operations between inputs $f : \mathbb{R} \to \mathbb{R}^{N_{in}}$ and convolutional kernels $\psi : \mathbb{R} \to \mathbb{R}^{N_{out} \times N_{in}}$ to produce outputs $(f * \psi) : \mathbb{R} \to \mathbb{R}^{N_{out}}$ as the linear combination along the $N_{in}$ dimension of convolutions with several learned convolutional kernels computed in parallel:

$$(f * \psi)_o = \sum_{i=1}^{N_{in}} (f_i * \psi_i), \quad o \in [1, 2, ..., N_{out}]. \qquad (24)$$

In practice, both lifting and group convolutional layers adhere to the same structure. In a dilation-translation convolutional layer with $N_{out}$ output channels, $N_{out}$ independent convolutional kernels, each consisting of $N_{in}$ channels, are learned. During the forward pass, the input is group-convolved with each of these kernels in parallel. The $N_{out}$ output channels are then formed by linearly combining the outcomes of the $N_{in}$ channels. In other words, lifting and group convolutional layers produce linear combinations of distinct time-frequency decompositions of the input computed in parallel at each layer.

**Wavelet networks are scale-translation equivariant nested non-linear time-frequency transforms.** Just like in conventional neural architectures, the outputs of lifting and group convolutional layers are interleaved with point-wise nonlinearities. Therefore, wavelet networks compute nonlinear scale-translation equivariant feature representations that resemble nested nonlinear time-frequency transforms of the input.

## 6 Experiments

In this section, we empirically evaluate wavelet networks. To this end, we take existing neural architectures designed to process raw signals and construct equivalent wavelet networks (W-Nets). We then compare the performance of W-Nets and the corresponding baselines on tasks defined on raw environmental sounds, raw audio and raw electric signals. We replicate as close as possible the training regime of the corresponding baselines and utilize their implementation as a baseline whenever possible. Detailed descriptions of the specific architectures as well as the hyperparameters used for each experiment are provided in Appx. C.

### 6.1 Classification of environmental sounds

First, we consider the task of classifying environmental sounds on the UrbanSound8K (US8K) dataset (Salamon et al., 2014). The US8K dataset consists of 8732 audio clips uniformly drawn from 10 environmental sounds, e.g., siren, jackhammer, etc, of 4 seconds or less, with a total of 9.7 hours of audio.

**Experimental setup.** We compare the M$n$-Nets of Dai et al. (2017) and the 1DCNNs of Abdoli et al. (2019) with equivalent W-Nets in terms of number of layers and parameters. Contrarily to Dai et al. (2017) we sample the audio files at 22.05kHz as opposed to 8kHz. This results from preliminary studies of the data, which indicated that some classes become indistinguishable for the human ear at such low sampling rates.[2] For the comparison with the 1DCNN of Abdoli et al. (2019), we select the 50999-1DCNN as baseline, as it is the network type that requires the less human engineering. We note, however, that we were unable to replicate the results reported in Abdoli et al. (2019). In contrast to the 83±1,3% reported, we were only able to obtain a final accuracy of 62.0±6.791. This inconsistency is further detailed in Appx. C.1.

To compare to models other than M$n$-nets and 1DCNNs, e.g., Pons et al. (2017a); Tokozume & Harada (2017), we also provide 10-fold cross-validation results. This is done by taking 8 of the 10 official subsets for

---

[2]See https://github.com/dwromero/wavelet_networks/blob/master/experiments/UrbanSound8K/data_analysis.ipynb.

Table 1: Experimental results on UrbanSound8K.

| UrbanSound8K | | | |
|---|---|---|---|
| Model | 10ᵀᴴ Fold Acc. (%) | Cross-Val. Acc. (%) | # Params. |
| M3-Net | 54.48 | - | 220.67к |
| W3-Net | 61.05 | - | |
| W3-Net-wl | **63.08** | - | 219.45к |
| M5-Net | 69.89 | - | 558.08к |
| W5-Net | 72.28 | - | |
| W5-Net-wl | **74.55** | - | 558.03к |
| M11-Net | 74.43 | - | 1.784м |
| W11-Net | 79.33 | 66.97 ± 5.178 | |
| W11-Net-wl | **80.41** | **68.47 ± 4.914** | 1.806м |
| M18-Net | 69.65 | - | 3.680м |
| W18-Net | 75.87 | 64.02 ± 4.645 | |
| W18-Net-wl | **78.26** | **65.01 ± 5.431** | 3.759м |
| M34-Net | 75.15 | - | 3.978м |
| W34-Net | 76.22 | 65.69 ± 5.780 | |
| W34-Net-wl | **78.38** | **66.77 ± 4.771** | 4.021м |
| 1DCNN | - | 62.00 ± 6.791 | 453.42к |
| W-1DCNN | - | 62.47 ± 4.925 | |
| W-1DCNN-wl | - | **62.64 ± 4.979** | 458.61к |

| Comparison With Other Approaches | | | |
|---|---|---|---|
| Model | Type | Cross-Val. Acc. (%) | # Params. |
| W11-Net-wl | Raw | 68.47 ± 4.914 | 1.806м |
| PiczakCNN Piczak (2015) | Mel Spectrogram | 73.7 | 26м |
| VGG Pons & Serra (2019) | | 70.74 | 77м |
| EnvNet-v2 Tokozume & Harada (2017) | Raw (Bagging) | **78** | 101м |

training, one for validation and one for test. We consistently select the $(n-1) \bmod 10$ subset for validation when testing on the $n$-th subset. We note that this training regime might be different from those used in other works, as previous works often do not disclose which subsets are used for validation.

**Results.** Our results (Tab. 1) show that wavelet networks consistently outperform CNNs on raw waveforms. In addition, they are competitive to spectrogram-based approaches, while using significantly fewer parameters and bypassing the need for preprocessing. Furthermore, we observe that encouraging wavelet structure to the convolutional kernels –denoted by the WL suffix– consistently leads to improved accuracy.

## 6.2   Automatic music tagging

Next, we consider the task of automatic music tagging on the MagnaTagATune (MTAT) dataset (Law et al., 2009). The MTAT dataset consists of 25879 audio clips with a total of 170 hours of audio, along with several per-song tags. The goal of the task is to provide the right tags to each of the songs in the dataset.

**Experimental setup.** Following Lee et al. (2017), we extract the most frequently used 50 tags and trim the audios to 29.1 seconds at a sample-rate of 22.05kHz. Following the convention in literature, we use ROC-curve (AUC) and mean average precision (MAP) as performance metrics. We compare the best performing model of Lee et al. (2017), the $3^9$-Net with a corresponding wavelet network denoted W3$^9$-Net.

**Results.** Our results (Tab. 2) show that wavelet networks consistently outperform CNNs on raw waveforms and perform competitively to spectrogram-based approaches in this dataset as well. In addition, we observe that encouraging the learning of wavelet-like kernels consistently results in increased accuracy as well.

## 6.3   Bearing fault detection

Finally, we also validate Wavelet networks for the task of condition monitoring in induction motors. To this end, we classify healthy and faulty bearings from raw data provided by Samotics. The dataset consists of 246 clips of 15 seconds sampled at 20kHz. The dataset is slightly unbalanced containing 155 healthy and 91 faulty recordings [155, 91]. The dataset is previously split into a training set of [85, 52] and a test

Table 2: Experimental results on MTAT.

| | MagnaTagATune | | | | |
|---|---|---|---|---|---|
| MODEL | AVERAGE AUC | | MAP | | # PARAMS. |
| | PER-CLASS | PER-CLIP | PER-CLASS | PER-CLIP | |
| $3^9$-NET | 0.893 | 0.936 | 0.385 | 0.700 | 2.394M |
| W$3^9$-NET | 0.895 | 0.941 | 0.397 | 0.719 | 2.404M |
| W$3^9$-NET-WL | **0.899** | __0.943__ | **0.404** | __0.723__ | |

| | Comparison With Other Approaches | | | | |
|---|---|---|---|---|---|
| MODEL | AVERAGE AUC | | MAP | | # PARAMS. |
| | PER-CLASS | PER-CLIP | PER-CLASS | PER-CLIP | |
| PCNN LIU ET AL. (2016) | 0.9013 | **0.9365** | __0.4267__ | 0.6902 | - |
| CNN PONS ET AL. (2017A)* (RAW) | 0.8905 | - | 0.3492 | - | 11.8M |
| CNN PONS ET AL. (2017A)* (SPECT.) | __0.9040__ | - | 0.3811 | - | 5M |
| CNN PONS ET AL. (2017B) (SPECT.) | 0.893 | - | - | - | 191K |

* Reported results are obtained in a more difficult version of this dataset.

Table 3: Experimental results on bearing fault detection.

| MODEL | ACC. (%) | # PARAMS. |
|---|---|---|
| M11-NET | 65.1376 | 1.806M |
| W11-NET | **68.8073** | 1.823M |
| W11-NET-WL | **70.207** | |

set of $[70, 39]$ samples, respectively. These splits are provided ensuring that measurements from the same motor are not included both in the train and the test set. We utilize 20% of the training set for validation. Each clip is composed of 6 channels measuring both current and voltage on the 3 poles of the motor.

**Experimental setup.** We take the best performing networks on the US8K dataset: the M-11 and W-11 networks, and utilize variants of these architectures for our experiments on this dataset.

**Results.** Once again we observe that Wavelet networks outperform CNNs on raw waveforms and encouraging the learning of wavelet-like kernels consistently improves accuracy (Tab. 3).

## 6.4 Discussion

Our empirical results firmly establish wavelet networks as a promising avenue for learning from raw time-series data. Notably, these results highlight that considering the symmetries inherent to time-series data –namely translation and scale– for the development of neural networks consistently leads to improved outcomes. Furthermore, we observe that the benefits of wavelet networks extend beyond sound and audio domains. This result advocates for the use of wavelet networks and scale-translation equivariance for learning on time-series data from different sources, e.g., financial data, sensory data. Finally, we also note that promoting the learning of wavelet-like convolutional kernels consistently leads to improved outcomes. We posit that this discovery may hold broader implications for group equivariant networks in general.

**Relation to scale-equivariant models of images and 2D signals.** In the past, multiple scale-equivariant models have been proposed for the processing of images and 2D signals (Worrall & Welling, 2019; Sosnovik et al., 2020; 2021). Interestingly, we find that the difference in the lengths of the inputs received by image and time-series models leads to very different insights per modality. For comparison, Sosnovik et al. (2020) considers images up to 96×96 pixels, whereas audio files in the US8K dataset are 32.000 samples long. We find that this difference in input lengths has crucial implications for how scale interactions within scale-equivariant models function. Sosnovik et al. (2020) mentions that using inter-scale interactions introduces additional equivariance errors due to the truncation of the set $\mathcal{S}$. Therefore, their networks are built with either no scale interaction or interactions of maximum 2 scales. This strongly contrasts with time-series where incorporating inter-scale interactions consistently leads to performance improvements. In our case, the number of scales and inter-scale interactions is rather constrained by the size and computational cost of convolutional kernels (Sec. 5.2.2) rather than their potential negative impact on the model's accuracy.

## 7  Limitations and future work

**Memory and time consumption grows proportionally to the number of scales considered.** The biggest limitation of our approach is the increase in memory and time demands as the number of scales considered grows. One potential avenue to mitigate this could involve adopting Monte-Carlo approximations for the computation of group convolutions (Finzi et al., 2020). This strategy might not only establish equivariance to the continuous scale group –in expectation–, but also dramatically reduce the number of scales considered in each forward pass. Another intriguing direction lies in the extension of partial equivariance (Romero & Lohit, 2022) to the scale group. This extension would enable learning the subset of scales to which the model is equivariant, which in turn could lead to faster execution and enhanced adaptability. Lastly, the adaptation of separable group convolutions (Knigge et al., 2022) offers a means to reduce the computational and memory requirements of wavelet networks.

**Convolutions with large convolutional kernels: parameterization and efficiency.** The foundation of our approach hinges on computing convolutions with banks of dilated convolutional kernels (Eq. 12, 11). Consequently, considering how these kernels are parameterized as well as how these convolutions are computed can unveil avenues for future improvement. Recently, Romero et al. (2021) introduced an expressive continuous parameterization for (large) convolutional kernels that has proven advantageous for complex tasks such as large language modelling (Poli et al., 2023) and processing DNA chains (Nguyen et al., 2023). Exploring the use of this parameterization for wavelet networks could lead to valuable insights and improvements, potentially surpassing the current utilization of $B^2$-spline bases. Furthermore, convolutional networks that rely on convolutions with very large convolutional kernels, e.g., Romero et al. (2021); Poli et al. (2023); Nguyen et al. (2023), leverage the Fourier transform to compute convolutions in the frequency domain. In the context of wavelet networks, dynamically selecting between spatial and Fourier convolutions based on the size of convolutional kernels has the potential to significantly improve their efficiency.

## 8  Conclusion

In conclusion, this study introduces *Wavelet Networks*, a new class of neural networks for raw time-series processing that harness the symmetries inherent to time-series data –scale and translation– for the construction of neural architectures that respect them. We observe a clear connection between the wavelet transform and scale-translation group convolutions, establishing a profound link between our approach and classical spectro-temporal analysis. In contrast to the usual approach, which uses spectro-temporal representations as a frontend for the subsequent use of 2D CNNs, wavelet networks consistently preserve these symmetries across the whole network through the use of convolutional layers that resemble the wavelet transform. Our analysis reveals that wavelet networks combine the benefits of wavelet-like time-frequency decompositions with the adaptability and non-linearity of neural networks.

Our empirical results demonstrate the superiority of Wavelet Networks over conventional CNNs on raw time-series data, achieving comparable performance to approaches that rely on engineered spectrogram-based methods, e.g., log-Mel spectrograms, with reduced parameters and no need for preprocessing.

This work pioneers the concept of scale-translation equivariant neural networks for time-series analysis, opening new avenues for time-series processing.

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

# Appendix

## A  Group and group action

**Group.** A group is an ordered pair $(\mathcal{G}, \cdot)$ where $\mathcal{G}$ is a set and $\cdot : \mathcal{G} \times \mathcal{G} \to \mathcal{G}$ is a binary operation on $\mathcal{G}$, such that *(i)* the set is closed under this operation, *(ii)* the operation is associative, i.e., $(g_1 \cdot g_2) \cdot g_3 = g_1 \cdot (g_2 \cdot g_3)$, $g_1, g_2, g_3 \in \mathcal{G}$, *(iii)* there exists an identity element $e \in \mathcal{G}$ such that $\forall g \in \mathcal{G}$ we have $e \cdot g = g \cdot e = g$, and *(iv)* for each $g \in \mathcal{G}$, there exists an inverse $g^{-1}$ such that $g \cdot g^{-1} = e$.

**Subgroup.** Given a group $(\mathcal{G}, \cdot)$, we say that a subset $\mathcal{H}$ is a subgroup of $\mathcal{G}$ if the tuple $(\mathcal{H}, \cdot)$ also complies to the group axioms. For example, the set of rotations by $90°$, $\mathcal{H} = \{0°, 90°, 180°, 270°\}$, is a subgroup of the continuous rotation group as it also complies to the group axioms.

**Group action.** Let $\mathcal{G}$ be a group and $\mathcal{X}$ be a set. The (left) group action of $\mathcal{G}$ on $\mathcal{X}$ is a function

$$\mathcal{A} : \mathcal{G} \times \mathcal{X} \to \mathcal{X}, \quad \mathcal{A}_g : x \to x', \tag{25}$$

such that for any $g_1$, $g_2 \in \mathcal{G}$, $\mathcal{A}_{g_2 g_1} = \mathcal{A}_{g_2} \circ \mathcal{A}_{g_1}$. In other words, the action of $\mathcal{G}$ on $\mathcal{X}$ describes how the elements in the set $x \in \mathcal{X}$ are transformed by elements $g \in \mathcal{G}$. For brevity, $\mathcal{A}_g(x)$ is written as $gx$.

## B  Equivariance properties of common time-frequency transforms

### B.1  The Fourier transform

The Fourier transform represents a function with finite energy $f \in \mathrm{L}^2(\mathbb{R})$ as a sum of complex sinusoidal waves $\mathrm{e}^{i\omega t} = \cos \omega t + i \sin \omega t$:

$$f(t) = \frac{1}{2\pi} \int_{-\infty}^{\infty} \hat{f}(\omega) \, \mathrm{e}^{i\omega t} \, \mathrm{d}\omega,$$

where, $\hat{f}(\omega)$ depicts the amplitude of each component $\mathrm{e}^{i\omega t}$ in $f$. The *Fourier transform* $\mathcal{F}$ is defined as:

$$\mathcal{F}[f](\omega) = \hat{f}(\omega) = \langle f, \mathrm{e}^{i\omega t} \rangle = \int_{-\infty}^{\infty} f(t) \, \mathrm{e}^{-i\omega t} \, \mathrm{d}t.$$

In other words, the Fourier transform encodes $f$ into a time-frequency dictionary $\mathcal{D} = \{\mathrm{e}^{i\omega t}\}_{\omega \in \mathbb{R}}$.

**Input translation.** Let $\mathcal{L}_{t_0}[f](t) = f(t - t_0)$ be a translated version of $f$. Its Fourier transform is given by:

$$\mathcal{F}[\mathcal{L}_{t_0}[f]](\omega) = \int_{-\infty}^{\infty} f(t - t_0) \, \mathrm{e}^{-i\omega t} \, \mathrm{d}t \, \Big| \, \tilde{t} = t - t_0; \, \mathrm{d}\tilde{t} = \mathrm{d}t$$

$$= \int_{-\infty}^{\infty} f(\tilde{t}) \, \mathrm{e}^{-i\omega(\tilde{t} + t_0)} \, \mathrm{d}\tilde{t} = \mathrm{e}^{-i\omega t_0} \int_{-\infty}^{\infty} f(\tilde{t}) \, \mathrm{e}^{-i\omega \tilde{t}} \, \mathrm{d}\tilde{t} = \mathrm{e}^{-i\omega t_0} \mathcal{F}[f](\omega) \tag{26}$$

In other words, a translation of $t_0$ corresponds to a phase modulation of $\mathrm{e}^{-i\omega t_0}$ in the frequency domain.

**Input scaling.** Let $\mathcal{L}_{s_0}[f](t) = f(s_0^{-1}t)$, $s_0 \in \mathbb{R}_{>0}$, be a scaled version of $f$. Its Fourier transform equals:

$$\mathcal{F}[\mathcal{L}_{s_0}[f]](\omega) = \int_{-\infty}^{\infty} f(s_0^{-1}t) \, \mathrm{e}^{-i\omega t} \, \mathrm{d}t \, \Big| \, \tilde{t} = s_0^{-1}t; \, \mathrm{d}\tilde{t} = s_0^{-1}\mathrm{d}t$$

$$= \int_{-\infty}^{\infty} f(\tilde{t}) \, \mathrm{e}^{-i\omega(s_0\tilde{t})} \, \mathrm{d}(s_0\tilde{t}) = s_0 \int_{-\infty}^{\infty} f(\tilde{t}) \, \mathrm{e}^{-i(s_0\omega)\tilde{t}} \, \mathrm{d}\tilde{t}$$

$$= s_0 \mathcal{F}[f](s_0\omega) = s_0 \mathcal{L}_{s_0^{-1}}[\mathcal{F}[f]](\omega) \tag{27}$$

In other words, we observe that a dilation on the time domain produces a compression in the Fourier domain times the inverse of the dilation.

**Simultaneous input translation and scaling.** Following the same derivation procedure, we can show the behavior of the Fourier transform to simultaneous translations and dilations of the input:

$$\mathcal{F}[\mathcal{L}_{s_0}\mathcal{L}_{t_0}[f]](\omega) = s_0 \mathrm{e}^{-i\omega t_0} \mathcal{F}[f](s_0\omega) = \mathrm{e}^{-i\omega t_0} s_0 \mathcal{L}_{s_0^{-1}}[\mathcal{F}[f]](\omega). \tag{28}$$

This corresponds to the superposition of the previously exhibited behaviours.

**Effect of input transformations on the spectral density.** The spectral density of a function $f \in L^2(\mathbb{R})$ is given by $|\mathcal{F}[f](\omega)|^2$. Input translations and dilations produce the following transformations:

$$|\mathcal{F}[\mathcal{L}_{t_0}[f]](\omega)|^2 = |\mathcal{F}[f](\omega)|^2 \tag{29}$$

$$|\mathcal{F}[\mathcal{L}_{s_0}[f]](\omega)|^2 = |s_0|^2 |\mathcal{L}_{s_0^{-1}}[\mathcal{F}[f]](\omega)|^2 \tag{30}$$

**Equivariance and invariance properties of the Fourier transform.** From Eq. 26 we can see that the Fourier transform is translation equivariant as it encodes translations of the input as a phase modulation of the output. In addition, it is also scale equivariant (Eq. 27), as it encodes dilations of the input as a modulation of the frequency components in the output. We can prove that the Fourier transform is dilation and translation equivariant by showing that the output transformations $e^{-i\omega t_0}$ and $s_0 \mathcal{L}_{s_0^{-1}}$ are group representations of the translation and scaling group in the Fourier space.

> **Group representation.** Let $\mathcal{G}$ be a group and $f$ be a function on a given functional space $L_V(\mathcal{X})$. The (left) regular representation of $\mathcal{G}$ is a linear transformation $\mathcal{L} : \mathcal{G} \times L_V(\mathcal{X}) \to L_V(\mathcal{X})$ which extends group actions to functions on $L_V(\mathcal{X})$ by:
>
> $$\mathcal{L}_g : f \to f', \quad f'(\mathcal{A}_g(x)) = f(x) \Leftrightarrow f'(x) = f(g^{-1}x),$$
>
> such that for any $g_1, g_2 \in \mathcal{G}$, $\mathcal{L}_{g_2 g_1} = \mathcal{L}_{g_2} \circ \mathcal{L}_{g_1}$. In other words, the group representation describes how a function on a functional space $f \in L_V(\mathcal{X})$ is modified by the effect of group elements $g \in \mathcal{G}$.

We can show that the combination of input translations $t_0, t_1 \in \mathbb{R}$ or dilations $s_0, s_1 \in \mathbb{R}_{>0}$ produces a transformation on the Fourier domain that preserves the group structure. In other words, that the transformations previously outlined are group representations. Specifically, for $\mathcal{L}_{t_1}[\mathcal{L}_{t_0}[f]]$ and $\mathcal{L}_{s_1}[\mathcal{L}_{s_0}[f]]$ it holds:

$$\mathcal{F}\big[\mathcal{L}_{t_1}[\mathcal{L}_{t_0}[f]]\big](\omega) = e^{-i\omega t_1}e^{-i\omega t_0}\mathcal{F}[f](\omega) = e^{-i\omega(t_1+t_0)}\mathcal{F}[f](\omega) = \mathcal{L}_{t_1+t_0}^{\text{Fourier}}[\mathcal{F}[f]](\omega)$$

$$\mathcal{F}\big[\mathcal{L}_{s_1}[\mathcal{L}_{s_0}[f]]\big](\omega) = s_1\mathcal{L}_{s_1^{-1}}\big[s_0\mathcal{L}_{s_0^{-1}}[\mathcal{F}[f]]\big](\omega) = (s_0 s_1)\mathcal{F}[f](s_1 s_0 \omega) = \mathcal{L}_{s_1 s_0}^{\text{Fourier}}[\mathcal{F}[f]](\omega)$$

with $\mathcal{L}_t^{\text{Fourier}}[\mathcal{F}[f]](\omega) = e^{-i\omega t}\mathcal{F}[f](\omega)$ the representation of the Fourier transform for the translation group, and $\mathcal{L}_s^{\text{Fourier}}[\mathcal{F}[f]](\omega) = s\mathcal{F}[f](s\omega)$ the representation of the Fourier transform for the dilation group.

Unfortunately, the resulting group representations rapidly become cumbersome specially in the presence of several input components. In addition, although the calculation of the spectral density leaves the scale equivariance property of the transformation unaffected, Eq. 29 shows that it reduces translation equivariance of the Fourier transform to *translation invariance*. This is why the Fourier transform is commonly considered not to carry positional information.

## B.2 The short-time Fourier transform

The short-time Fourier transform of a signal $f \in L^2(\mathbb{R})$ is given by:

$$\mathcal{S}[f](t, \omega) = \int_{-\infty}^{+\infty} f(\tau) w(\tau - t)\, e^{-i\omega\tau}\, d\tau.$$

In other words, it encodes the input $f$ into a time-frequency dictionary $\mathcal{D} = \{\phi_{t,\omega}\}$, $\phi_{t,\omega} = w(\tau - t)\, e^{-i\omega\tau}$.

**Input translation.** Let $\mathcal{L}_{t_0}[f](\tau) = f(\tau - t_0)$ be a translated version of $f$. Its short-time Fourier transform is given by:

$$\mathcal{S}[\mathcal{L}_{t_0}[f]](t, \omega) = \int_{-\infty}^{\infty} f(\tau - t_0) w(\tau - t)\, e^{-i\omega\tau}\, d\tau \ \bigg|\ \tilde{t} = \tau - t_0;\ d\tilde{t} = d\tau$$

$$= \int_{-\infty}^{\infty} f(\tilde{t}) w(\tilde{t} + t_0 - t)\, e^{-i\xi(\tilde{t}+t_0)}\, d\tilde{t} = e^{-i\xi t_0} \int_{-\infty}^{\infty} f(\tilde{t}) w(\tau - (t - t_0))\, e^{-i\omega\tilde{t}}\, d\tilde{t}$$

$$= e^{-i\omega t_0}\mathcal{S}[f](t - t_0, \omega) = e^{-i\omega t_0}\mathcal{L}_{t_0}[\mathcal{S}[f]](t, \omega) \tag{31}$$

In other words, a translation by $t_0$ in the time domain, corresponds to a shift by $t_0$ on the time axis of the short-time Fourier transform, and an additional phase modulation of $e^{-i\xi t_0}$ similar to that of the Fourier transform (Eq. 26).

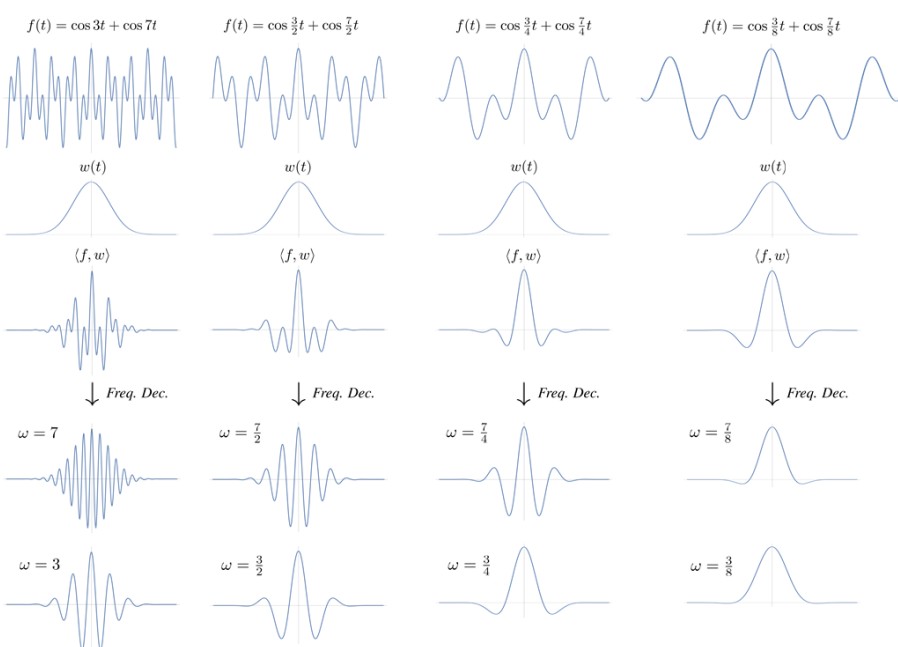

Figure 10: Scale equivariance of the short-time Fourier transform. Consider a function $f(t) = \cos \omega_1 t + \cos \omega_2 t$ composed of two frequencies $\omega_1 = 3$ and $\omega_2 = 7$, and a window function $w(t)$, with which the short-time Fourier transform is calculated. For relatively high frequencies (left column), the dot-product of $f$ and $w$, $\langle f, w \rangle$, is able to capture sufficient spectral information from $f$ to correctly extract the frequencies $\omega_1, \omega_2$ from it. However, for dilated versions of the same signal $f$ (right columns) obtained by reducing the frequency of the spectral components $\omega_1, \omega_2$ of $f$, the capacity of the dot-product $\langle f, w \rangle$ to capture the spectral information in the input gradually degrades and, eventually, is entirely lost. Consequently, scale equivariance holds (approximately) for scales for which *all* of the spectral components of the signal $f$ lie within the range of the window $w$.

**Input scaling.** Let $\mathcal{L}_{s_0}[f](\tau) = f(s_0^{-1}\tau)$, $s_0 \in \mathbb{R}_{>0}$, be a scaled version of $f$. Its short-time Fourier transform is given by:

$$\mathcal{S}[\mathcal{L}_{s_0}[f]](t, \omega) = \int_{-\infty}^{\infty} f(s_0^{-1}t) w(\tau - t) \, \mathrm{e}^{-i\omega\tau} \, \mathrm{d}\tau \; \Big| \; \tilde{t} = s_0^{-1}\tau; \; \mathrm{d}\tilde{t} = s_0^{-1}\mathrm{d}\tau$$

$$= \int_{-\infty}^{\infty} f(\tilde{t}) w(s_0\tilde{t} - t) \, \mathrm{e}^{-i\omega(s_0\tilde{t})} \, \mathrm{d}(s_0\tilde{t}) = s_0 \int_{-\infty}^{\infty} f(\tilde{t}) w(s_0\tilde{t} - t) \, \mathrm{e}^{-i(s_0\omega)\tilde{t}} \, \mathrm{d}\tilde{t} \; \Big| \; t = s_0^{-1}s_0 t$$

$$= s_0 \int_{-\infty}^{\infty} f(\tilde{t}) w(s_0(\tilde{t} - s_0^{-1}t)) \, \mathrm{e}^{-i(s_0\omega)\tilde{t}} \, \mathrm{d}\tilde{t} \; \Big| \; w(s\tau) \approx w(\tau)$$

$$\approx s_0 \int_{-\infty}^{\infty} f(\tilde{t}) w(\tilde{t} - s_0^{-1}t) \, \mathrm{e}^{-i(s_0\omega)\tilde{t}} \, \mathrm{d}\tilde{t} \approx s_0 \, \mathcal{S}[f](s_0^{-1}t, s_0\omega) \tag{32}$$

In other words, a dilation in the time domain produces a compression in the frequency domain analogous to the Fourier transform (Eq. 27). However, it is important to note that we rely on the approximate $w(x) \approx w(sx)$ to arrive to the final expression. Nevertheless, it is important to note that this approximate *does not generally holds in practice*. This approximation implies that the window function $w$ is invariant to scaling, which holds only for increasing window sizes, i.e., when the short-term Fourier transform starts to approximate the (global) Fourier transform.

**Simultaneous input translation and scaling.** Following the same derivation procedure, we can show the behavior of the short-time Fourier transform to simultaneous translations and scaling. We have that:

$$\mathcal{S}[\mathcal{L}_{s_0}\mathcal{L}_{t_0}[f]](t, \omega) = s_0\mathrm{e}^{-i\omega t_0}\mathcal{S}[f](s_0^{-1}(t - t_0), s_0\omega) \tag{33}$$

**Effect of input transformations on the spectrogram.** The spectrogram of a function $f \in \mathrm{L}^2(\mathbb{R})$ is given by $|\mathcal{S}[f](t,\omega)|^2$. Input translations and dilations produce the following transformations:

$$|\mathcal{F}[\mathcal{L}_{t_0}[f]](t,\omega)|^2 = |\mathcal{L}_{t_0}[\mathcal{S}[f]](t,\omega)|^2 \tag{34}$$

$$|\mathcal{F}[\mathcal{L}_{s_0}[f]](t,\omega)|^2 = |s_0|^2 |\mathcal{S}[f](s_0^{-1}t, s_0\omega)|^2 \tag{35}$$

**Equivariance and invariance properties of the short-time Fourier transform.** The short-time Fourier transform is *approximately* translation and scale equivariance in a manner similar to that of the Fourier transform. In contrast to the Fourier transform, however, it decomposes input translations into a translation $t - t_0$ and a phase shift $\mathrm{e}^{-i\omega t_0}$ in the output (Eq. 31). This decomposition can be interpreted as a rough estimate $t - t_0$ signalizing the position in which the window $w$ is localized, and a fine grained localization within that window given by the phase shift $\mathrm{e}^{-i\omega t_0}$ indicating the relative position of the pattern within the window $\mathcal{L}_{(t-t_0)}[w](\tau)$.

Equivariance to dilations is analogous to the Fourier transform up to the fact that time and frequency are now jointly described. However, since the window itself does not scale with the sampled frequency –as is the case in wavelet transforms–, exact equivariance is not obtained. Note that equivariance to dilations is only approximate, and is restricted to the set of scales that can be detected with the width of the window used (see Fig. 10 for a visual explanation). Since this is not generally the case, the short-time Fourier transform is *not scale equivariant.*

The calculation of the spectrogram leaves the scale equivariance property of the transformation unaffected and is equivalent in a join manner to the scale equivariance property of the Fourier transform (Eq. 32). Differently however, Eq. 34 shows that translation equivariance is partially preserved and only information about the phase shift within the window is lost. This is why the short-time Fourier transform is said to carry positional information, i.e., to be (approximately) translation equivariant.

### B.3 The Wavelet Transform

The wavelet transform of a signal $f \in \mathrm{L}^2(\mathbb{R})$ is given by:

$$\mathcal{W}[f](t,s) = \langle f, \psi_{t,s} \rangle = \int_{-\infty}^{+\infty} f(\tau) \frac{1}{\sqrt{s}} \psi^*\left(\frac{\tau - t}{s}\right) \mathrm{d}\tau,$$

and is equivalent to encoding $f$ into a time-frequency dictionary $\mathcal{D} = \{\psi_{t,s}\}_{u\in\mathbb{R}, s\in\mathbb{R}_{>0}}$, $\psi_{t,s}(\tau) = \frac{1}{\sqrt{s}}\psi^*\left(\frac{\tau-t}{s}\right)$.

**Input translation.** Let $\mathcal{L}_{t_0}[f](\tau) = f(\tau - t_0)$ be a translated version of $f$. Its wavelet transform is given by:

$$\mathcal{W}[\mathcal{L}_{t_0}[f]](t,s) = \int_{-\infty}^{\infty} f(\tau - t_0)\sqrt{s}^{-1}\psi^*\left(s^{-1}(\tau - t)\right) \mathrm{d}\tau \;\Big|\; \tilde{t} = \tau - t_0;\; \mathrm{d}\tilde{t} = \mathrm{d}\tau$$

$$= \int_{-\infty}^{\infty} f(\tilde{t})\sqrt{s}^{-1}\psi^*\left(s^{-1}(\tilde{t} + t_0 - t)\right) \mathrm{d}\tilde{t} = \int_{-\infty}^{\infty} f(\tilde{t})\sqrt{s}^{-1}\psi^*\left(s^{-1}(\tilde{t} - (t - t_0))\right) \mathrm{d}\tilde{t}$$

$$= \mathcal{W}[f](t - t_0, s) = \mathcal{L}_{t_0}\mathcal{W}[f](t,s) \tag{36}$$

In other words, a translation of the input produces an equivalent translation in the wavelet domain.

**Input scaling.** Let $\mathcal{L}_{s_0}[f](t) = f(s_0^{-1}t)$ be a scaled version of $f$. The corresponding wavelet transform is:

$$\mathcal{W}[\mathcal{L}_{s_0}[f]](t,s) = \int_{-\infty}^{\infty} f(s_0^{-1}t)\sqrt{s}^{-1}\psi^*\left(s^{-1}(\tau - t)\right) \mathrm{d}\tau \;\Big|\; \tilde{t} = s_0^{-1}\tau;\; \mathrm{d}\tilde{t} = s_0^{-1}\mathrm{d}\tau$$

$$= \int_{-\infty}^{\infty} f(\tilde{t})\sqrt{s}^{-1}\psi^*\left(s^{-1}(s_0\tilde{t} - t)\right) \mathrm{d}(s_0\tilde{t}) \;\Big|\; t = s_0^{-1}s_0 t$$

$$= \int_{-\infty}^{\infty} f(\tilde{t})\sqrt{s}^{-1}s_0\psi^*\left(s^{-1}s_0(\tilde{t} - s_0^{-1}t)\right) \mathrm{d}\tilde{t} \;\Big|\; s_0 = \sqrt{s_0^{-1}s_0^{-1}}^{-1}$$

$$= \sqrt{s_0} \int_{-\infty}^{\infty} f(\tilde{t})\sqrt{s_0^{-1}s}^{-1}\psi^*\left((s_0^{-1}s)^{-1}(\tilde{t} - s_0^{-1}t)\right) \mathrm{d}\tilde{t}$$

$$= \sqrt{s_0}\,\mathcal{W}[f](s_0^{-1}t, s_0^{-1}s) = \sqrt{s_0}\,\mathcal{L}_{s_0}\mathcal{W}[f](t,s) \tag{37}$$

In other words, a dilation $s_0$ in the input domain produces an *equivalent* dilation in the wavelet domain on both components $(t, s)$, multiplied by a factor $\sqrt{s_0}$. That is, the wavelet transform is translation equivariant.

**Simultaneous input translation and scaling.** Following the same procedure, we can show the behavior of the wavelet transform to simultaneous translations and dilations of the input:

$$\mathcal{W}[f(s_0^{-1}(\tau - t_0)](t, s) = \sqrt{s_0}\, \mathcal{W}[f](s_0^{-1}(t - t_0), s_0^{-1}s) = \sqrt{s_0}\, \mathcal{L}_{t_0}\mathcal{L}_{s_0}\mathcal{W}[f](t, s) \tag{38}$$

We observe that the Wavelet transform is the only time-frequency transform that respects equivariance with equivalent group representations in the input and output domain.

**Effect of input transformations on the scalogram.** The scalogram of a function $f \in \mathrm{L}^2(\mathbb{R})$ is given by $|\mathcal{W}[f](u, s)|^2$. Input translations and dilations produce the following transformations on the scalogram:

$$|\mathcal{W}[\mathcal{L}_{t_0}[f]](u, s)|^2 = |\mathcal{L}_{t_0}[\mathcal{W}[f]](u, s)|^2 \tag{39}$$

$$|\mathcal{W}[\mathcal{L}_{s_0}[f]](u, s)|^2 = |\mathcal{L}_{s_0}[\mathcal{W}[f]](u, s)|^2 \tag{40}$$

In other words the scalogram is exactly equivariant to both translations and dilations.

**Equivariance and invariance properties of the wavelet transform.** From Eq. 36, we can see that the wavelet transform is *exactly equivariant to translations* and the group representation of the output space equals that of the input space. Furthermore, translation equivariance is preserved in the scalogram as well (Eq. 39). Similarly, scale equivariance is preserved on the wavelet transform up to a multiplicative factor (Eq. 37). However, the scalogram preserves both translation and dilation equivariance exactly (Eq. 40).

We emphasize that the group representation on the output space resembles that of the input space. This behavior leads to much more straightforward group representations than that exhibited by the Fourier transform and the short-time Fourier transform. Additionally, exact scale equivariance is only obtained on the scalogram (Eq. 40), whilst for the wavelet transform it is retained up to multiplicative factor (Eq. 37). This behavior elucidates the fact that time-frequency transforms have been optimized for energy density representations rather than for the time-frequency representations themselves.

## C Experimental details

Whenever possible, we use existing code for the baselines of our wavelet networks as a starting point for the general infrastructure of our model. Specifically, we utilize the PyTorch implementation provided in https://github.com/philipperemy/very-deep-convnets-raw-waveforms and https://github.com/kyungyunlee/sampleCNN-pytorch as baseline for the US8K experiments and the MTAT experiments Lee et al. (2017), respectively. By doing so, we aim to preserve the reproducibility of the experiments in the baseline papers during our own experiments, as some important training factors are not specified in the baseline papers, e.g., the learning rate used in Dai et al. (2017). Unfortunately, Abdoli et al. (2019) do not provide code and we were forced to interpret some of the ambiguities in the paper, e.g., the pooling type utilized in the pooling layers and the loss metric used.

Any omitted parameters can safely be considered to be the default values in `PyTorch 1.5.0`. Our experiments are carried out in a Nvidia TITAN RTX GPU.

### C.1 UrbanSound8K

**W$n$-Nets.** We use a sampling rate of 22.05kHz as opposed to the 8kHz used in Dai et al. (2017). An early study that indicated that some classes were indistinguishable for the human ear at this sampling rate.[3] We zero-pad signals shorter than 4 seconds so that all input signals have a constant length of 80200 samples. Following the implementation of Dai et al. (2017), we utilize the Adam optimizer (Kingma & Ba, 2014) with `lr=1e-2` and `weight_decay=1e-4`, and perform training on the official first 9 folds and test on the $10^{th}$ fold. We noticed that reducing the learning rate from `1e-2` to `1e-3` increased the performance of our W-Nets. The reported results of the W-Net variants are obtained with this learning rate.

We utilize batches of size 16 and perform training for 400 epochs. The learning rate is reduced by half after 20 epochs of no improvement in validation loss. The W$n$-nets used are specified in Table 4. See Dai et al. (2017, Tab. 1) for comparison.

---

[3]See https://github.com/dwromero/wavelet_networks/experiments/UrbanSound8K/data_analysis.ipynb.

Table 4: W$n$-networks. W3-Net (0.219M) denotes a 3-layer network with 0.219M parameters. $[79/4, 150, 3]$ denotes a group convolutional layer with a nominal kernel size of 79 samples, 150 filters and 3 scales, with a stride of 4. Stride is omitted for stride 1 (e.g., $[3, 150, 3]$ has stride 1). Each convolutional layer uses batch normalization right after the convolution, after which ReLU is applied. Following the findings of Romero et al. (Romero et al., 2020, Appx. C) on the influence of stride in the equivariance of the network, we replace strided convolutions by normal convolutions, followed by spatial pooling. $[\dots] \times k$ denotes $k$ stacked layers and double layers in brackets denote residual blocks as defined in (Dai et al., 2017, Fig. 1b). In each of the levels of convolutional layers and residual blocks, the first convolution of the first block has scale 3 and the remaining convolutional layers at that level has scale 1.

| W3-Net (0.219M) | W5-Net (0.558M) | W11-Net (1.806M) | W18-Net (3.759M) | W34-Net (4.021M) |
|---|---|---|---|---|
| INPUT: 80200x 1 TIME-DOMAIN WAVEFORM | | | | |
| LIFTING LAYER ( 9 SCALES) | | | | |
| $[79/4, 150]$ | $[79/4, 74]$ | $[79/4, 51]$ | $[79/4, 57]$ | $[79/4, 45]$ |
| MAXPOOL: 4x1 (OUTPUT: 80200x 9x 1) | | | | |
| $[3, 150, 3]$ | $[3, 74, 3]$ | $[3, 51, 3] \times 2$ | $[3, 57, 3] \times 4$ | $\begin{bmatrix} 3, 45 \\ 3, 45 \end{bmatrix} \times 3$ |
| MAXPOOL: 4x1 (OUTPUT: 80200x 7x 1) | | | | |
| | $[3, 148, 3]$ | $[3, 102, 3] \times 2$ | $[3, 114, 3] \times 4$ | $\begin{bmatrix} 3, 90 \\ 3, 90 \end{bmatrix} \times 4$ |
| MAXPOOL: 4x1 (OUTPUT: 80200x 5x 1) | | | | |
| | $[3, 296, 3]$ | $[3, 204, 3] \times 3$ | $[3, 228, 3] \times 4$ | $\begin{bmatrix} 3, 180 \\ 3, 180 \end{bmatrix} \times 6$ |
| MAXPOOL: 4x1 (OUTPUT: 80200x 3x 1) | | | | |
| | | $[3, 408, 3] \times 2$ | $[3, 456, 3] \times 4$ | $\begin{bmatrix} 3, 360 \\ 3, 360 \end{bmatrix} \times 3$ |
| GLOBAL AVERAGE POOLING (OUTPUT: 1 X N) | | | | |
| SOFTMAX [110] (OUTPUT: 1 X N) | | | | |

Table 5: Wavelet network variant of the 50999-1DCNN (Abdoli et al., 2019). $[31/2, 24, 3]$ denotes a group convolutional layer with a nominal kernel size of 31 samples, 24 filters and 3 scales, with a stride of 2. FC: $[96, 48]$ denotes a fully-connected layer with 96 input channels and 48 output channels. Each convolutional layer uses batch normalization right after the convolution followed by ReLU. All fully connected layers expect for the last one use dropout of 0.25 and ReLU. Following the findings of Romero et al. (2020, Appx. C) on the influence of stride in the equivariance of the network, we replace strided convolutions with normal convolutions followed by spatial pooling. We note that the input size of our network is (presumably) different from that in Abdoli et al. (2019). Consequently, the last pooling layer utilizes a region of 5, in contrast to 4 as used in Abdoli et al. (2019). However, as it is not clear how the input dimension is reduced from 64000 to 50999 in Abdoli et al. (2019) and we stick to their original sampling procedure. We interpret their poling layers as max-pooling ones.

| W-1DCNN (0.549M) |
| :---: |
| INPUT: 64000x 1 |
| *LIFTING LAYER ( 9 SCALES)* |
| $[63/2, 12]$ |
| MAXPOOL: 8x1 |
| $[31/2, 24, 3]$ |
| MAXPOOL: 8x1 |
| $[15/2, 48, 3]$
$[7/2, 96, 3]$
$[3/2, 408, 3]$ |
| MAXPOOL: 5x1 |
| FLATTEN $196 \times 6 \rightarrow 1152$
FC: $[1152, 96]$
FC: $[96, 48]$
FC: $[48, 10]$ |
| SOFTMAX |

**W-1DCNN.** Following Abdoli et al. (2019), we utilize a sampling rate of 16kHz during our experiments. We zero-pad signals shorter than 4 seconds so that all input signals have a constant length of 64000 samples. Following the experimental description of the paper, we utilize the AdaDelta optimizer (Zeiler, 2012) with `lr=1.0` and perform training in a 10-fold cross validation setting as described in Sec. 6. We use batches of size 100 and perform training for 100 epochs. We utilize the 50999-1DCNN variant of Abdoli et al. (2019), as it is the variant that requires the less human engineering.[4]

Unfortunately, we were not able to replicate the results reported in Abdoli et al. (2019) (83±1.3%) in our experiments. Our replication of Abdoli et al. (2019) lead to a 10-cross fold accuracy of 62±1.3%, which is 21 accuracy points less relative to the results reported. We experiment with our interpretation of the mean squared logarithmic error (MSLE) loss defined in (Abdoli et al., 2019, Eq. 4). However, we find that the conventional cross-entropy loss leads to better results. Consequently, all our reported results are based on training with this loss.[5]

The description of the Wavelet 50999-1DCNN Abdoli et al. (2019) is provided in Table 5 (see (Abdoli et al., 2019, Tab. 1) for comparison).

---

[4]The remaining architectures partition the input signal into overlapping windows after which the predictions of each windows are summarized via a voting mechanism. Consequently, one could argue that the 50999-1DCNN is the only variant that truly receives the raw waveform signal. Nevertheless it is not clear from the paper how the input signal of 64000 samples is reduced to 50999 samples, which is the input dimension of the raw signal for this architecture type.

[5]The MSLE loss in Abdoli et al. (2019, Eq. 4) is defined as $\frac{1}{N} \sum_{i=1}^{N} \log \frac{p_i+1}{a_i+1}^2$, where $p_i$, $a_i$ and $N$ are the predicted class, the actual class, and the number of samples respectively. Note, however, that obtaining the predicted class $p_i$, i.e., $p_i = \arg\max_o f(x_i)$, where $f(x_i) \in \mathbb{R}^O$ is the output of the network for a classification problem with $O$ classes and input $x_i$, is a non-differentiable function. Consequently, it is not possible to train the network based on the formulation provided there. In order to train our model with this loss, we re-formulate the MSLE loss as $\frac{1}{N} \sum_{i=1}^{N} \sum_{o=1}^{O} \log \frac{p_{i,o}+1}{a_{i,o}+1}^2$, where $\{a_{i,o}\}_{o=1}^{O}$ is a one-hot encoded version of the label $a_i$. That is, we measure the difference between the one-hot encoded label and the output.

Table 6: W3$^9$-network. $[3/1, 90, 3]$ denotes a group convolutional layer with a nominal kernel size of 3 samples, 90 filters and 3 scales, with a stride of 1. MP:3x1 denotes a max-pooling layer of size 3. FC: $[360, 50]$ denotes a fully-connected layer with 360 input channels and 50 output channels. Each convolutional layer uses batch normalization after the convolution followed by ReLU. Dropout of 0.5 is used after the $6^{th}$ and $11^{th}$ layer. Following the findings of Romero et al. (2020, Appx. C) on the influence of stride in the network equivariances, we replace strided convolutions by normal convolutions followed by spatial pooling.

| W-3$^9$ Net (2.404m) |
| :---: |
| Input: 59049x 1 |
| *Lifting Layer ( 9 scales)* |
| $[3/3, 90]$ |
| $[3/1, 90, 3]$, MP:3x1 |
| $[3/1, 90, 1]$, MP:3x1 |
| $[3/1, 180, 1]$, MP:3x1 |
| $[3/1, 180, 3]$, MP:3x1 |
| $[3/1, 180, 1]$, MP:3x1 |
| $[3/1, 180, 1]$, MP:3x1 |
| $[3/1, 180, 3]$, MP:3x1 |
| $[3/1, 180, 1]$, MP:3x1 |
| $[3/1, 360, 1]$, MP:3x1 |
| $[3/1, 360, 3]$ |
| FC: $[360, 50]$ |
| Sigmoid |

## C.2    MagnaTagATune

**W3$^9$-Network.** For the experiments in the MTAT dataset, we utilize the PyTorch code provided by Lee et al. Lee et al. (2017). We use the data and tag preprocessing used in Lee et al. (2017). We utilize the SGD optimizer with `lr=1e-2`, `weight_decay=1e-6` and `nesterov=True`. We use batches of size 23 and perform training for 100 epochs. The learning rate is reduced by 5 after 3 epochs of no improvement in the validation loss. Early stopping is used if the learning rate drops under `1e-7`.

We were unable to replicate the per-class AUC results reported in Lee et al. (2017). Our experiments indicated a per-class AUC of 0.893 instead of the 0.905 reported in Lee et al. (2017). Details of the W3$^9$-Net used are given in Table 6 (see (Lee et al., 2017, Tab. 1) for comparison).

