# OpenReview forum: "Wavelet Networks: Scale-Translation Equivariant Learning From Raw Time-Series"
_TMLR — Accepted by TMLR_

### Review · Reviewer_R4e9 · 2023-10-10

**Summary Of Contributions:**

The paper introduces the wavelet network, an equivariant network tailored for 1D signals that ensures scaling and translation equivariance. The authors explained the connection between the (lifting) group convolution and the wavelet transform. Numerical experiments are presented to demonstrate the improved performance of the proposed model compared to conventional models, including CNNs.

**Audience:**

No

**Claims And Evidence:**

No

**Requested Changes:**

Please see the previous section

**Strengths And Weaknesses:**

**Weakness**
1. The proposed model is essentially a special case of G-CNN (with regular representations) for the scaling-translation group acting on 1d signals. More specifically, the first layer is the lifting group convolution, and all subsequent layers are the scaling-translation group convolutions. This model structure is not novel; it has been extensively explored and documented in prior literature. Consequently, it appears that the authors' contribution in this domain does not introduce any new concepts.

2. References [1, 2] have proposed a remarkably similar architecture targeting this group, albeit for 2D signals. Regrettably, there is an apparent omission in the current manuscript, as these two prior works have not been acknowledged.

3. The proposition of using continuous bases is neither groundbreaking nor original. Both references [1] and [2] have previously used this approach to mitigate interpolation artifacts.

4. The numerical experiments are conducted to compare only with non-equivariant models. A more compelling evaluation would involve comparisons against, for instance, the model in [1] when adapted for 1D signals.

**Strength**
1. The paper is well-written. The regularization on "wavelet structure" is interesting. However, it is hard to understand why this regularization helps the overall generalization performance.

[1] Sosnovik, Ivan, Michał Szmaja, and Arnold Smeulders. "Scale-Equivariant Steerable Networks." International Conference on Learning Representations. 2019.
[2] Zhu, Wei, et al. "Scaling-translation-equivariant networks with decomposed convolutional filters." The Journal of Machine Learning Research 23.1 (2022): 2958-3002.

---

> ### Author Response · Authors · 2023-10-11
>
> Dear reviewer R4e9,
>
> Thank you very much for review. We sincerely appreciate the time you spend in evaluating our work.
>
> Here, we respond to your observations:
>
> * **"The proposed model is essentially a special case of G-CNN ..."**
>
> This is true. However, as specified in the paper, the core of our contribution is the exploration of equivariant learning on time-series data. Based on our analysis, we find translation and scale to be important for time-series processing and explore this idea thoroughly --which to the best of our knowledge has not been done previously.
>
> To accentuate this fact, we have made some modifications to the document:
> 1. First, we have changed the title of the submission to Scale-Translation Equivariant Learning from Raw Time-Series to accentuate the fact that our study is focussed on time series learning.
> 2. We explicitly mention that scale equivariance has been proposed previously for 2D data in the inroduction by saying:
> > *"By leveraging group convolutions equivariant to the scale-translation group, we construct neural architectures such that when the input undergoes translation, scaling or a combination of the two, all intermediate layers will undergo an equal transformation in a hierarchical manner, akin to the methods proposed by Sosnovik et al. (2020); Zhu et al. (2022) for visual data."*
>
> * **References [1, 2] have proposed a remarkably similar architecture targeting this group, albeit for 2D signals...**
>
> We apologize for this unintended omission. We indeed had referenced [1] in a previous version of the manuscript, and forgot to add this reference back. We have now included both your proposed references, as well as Worrall and Welling (2019).
>
> * **The numerical experiments are conducted to compare only with non-equivariant models. A more compelling evaluation would involve comparisons against, for instance, the model in [1] when adapted for 1D signals.**
>
> To the best of our knowledge, there are no existing scale translation equivariant models on 1D. We therefore have not performed this comparison. Note that, our contribution is not to say that we construct a better scale-equivariant model than those in [1,2], but rather to explore the use of symmetry preservation by means of scale-translation equivariance for time-series processing. We therefore did not consider such comparisons.
>
>  * **The regularization on "wavelet structure" is interesting. However, it is hard to understand why this regularization helps the overall generalization performance.**
>
> The reason this leads to better results is explicitly stated in Sec. 5.2.3, and can be explained based on spectro-temporal theory. In short, since, in the eyes of spectro-temporal theory, scale-translation group convolutions is providing an spectro-temporal decomposition of the input, imposing a band-passing structure to the convolutional kernels --such that it aligns better with the design of Wavelets-- leads to cleaner spectral decompositions.
>
> ---
>
> We hope that these responses clarify your questions and concerns. Please let us know if you have any follow-up / additional questions.
>
> Best regards,
>
> The Authors

---

> > ### Comment · Reviewer_R4e9 · 2023-10-19
> > **Thank you for the detailed response**
> >
> > I thank the authors for the detailed response. I acknowledge that scaling-translation equivariant networks have not been proposed or studied specifically for time-series data. In this aspect, this paper is different from [1,2]. However, I also want to point out that the extension of [1,2] to 1d signals is not that novel/significant--the filters are all $L^1$-normalized, which resembles wavelet transforms (that are $L^2$ normalized).
> >
> > That being said, the experiments on 1d signals are indeed new.

---

> > > ### Author Response · Authors · 2023-10-26
> > >
> > > Dear reviewer,
> > >
> > > Thank you very much for your additional comment.
> > >
> > > We agree that other scale-equivariant architectures could be extended to 1D as well. However, please note that the paper provides other interesting insights --both theoretical and experimental-- with regard to how scale equivariant networks process their input, which to the best of our knowledge is novel. In addition, we analyze the suitability of other spectro-temporal representations and explore their equivariance properties: an analysis we consider important, as these transformations have been extensively used in practice across different settings.
> > >
> > > Please let us know if you have any remaining points of discussion.
> > >
> > > Best,
> > >
> > > The Authors.

---

### Review · Reviewer_47NA · 2023-10-10

**Summary Of Contributions:**

- This paper identifies two important symmetries inherent to time-serious data: scale and translation. Based on this,  the paper proposes scale-translation equivariant neural networks (named Wavelet Networks) for time-series data, which is largely underexplored in the literature on geometric deep learning.
- The authors discuss the relationship between the construction of scale-translation equivariant mappings in this paper with the well-known wavelet transform, thereby shedding some light on the merit of the proposed methods.
- Their empirical results demonstrate that the proposed Wavelet Networks outperform baselines such as conventional CNNs on a variety of tasks and time-serious types, e.g. audio, environmental sounds, and electrical signals.

**Audience:**

Yes

**Broader Impact Concerns:**

I have no concerns over the ethical implications.

**Claims And Evidence:**

Yes

**Requested Changes:**

In fact, I think this work is ready for publication and I don't have any essential requested changes.

**Strengths And Weaknesses:**

**Strengths**
- The paper tackles an important problem by exploring equivariant neural networks w.r.t. to scale and translation for time series data. Given the prevalence of time-series data across various applications and the demonstrated benefits of equivariant models in enhancing sample efficiency and generalization, this research is a valuable contribution.
- Instead of solely focusing on their methodologies, the authors offer an in-depth comparison with established techniques in the field, such as the wavelet transform and short-time Fourier transform. This comparative analysis sheds light on the reasons behind the observed performance enhancements. The discussions, including those on the differences between visual and spectro-temporal representations and between the short-term Fourier transform from the wavelet transform, are particularly interesting.
- The paper also delves into the practical aspects of their research, emphasizing considerations like discretization and computational complexity.

---

> ### Author Response · Authors · 2023-10-11
>
> Dear reviewer 47NA,
>
> Thank you very much for review. We sincerely appreciate the time you spend in evaluating our work.
>
> Best regards,
>
> The authors

---

### Review · Reviewer_A2P8 · 2023-10-11

**Summary Of Contributions:**

The paper proposes a wavelet network for scale-translation equivariant learning. The primary application area is time-series data, and the empirical verifications focus on sound signals. To my best knowledge, the group convolution and lifting convolution in the wavelet network are novel and serve the purpose of encoding scale-translation equivariance. The major contribution is to develop implementation details of the group convolution and lifting convolution, as well as testify their performance on real data.

**Audience:**

Yes

**Claims And Evidence:**

Yes

**Requested Changes:**

Section 4 might need a different highlighting style. Currently, the definition of Fourier transform and wavelet transform are highlighted in a grey box. However, from my understanding, the purpose of this section is to show the unique challenge and the insufficiency of existing ideas for handling auditory data. It could be better to articulate these problems in a more outstanding manner.

Section 5 looks very complicated. It begins with definitions of group convolutions, then talks about the wavelet network architecture, and lastly provides implementation details. I feel like a better presentation structure can be used, such as starting with the network architecture and then introduce building blocks in the network. The implementation details may deserve a separated section.

I am not an expert on auditory data processing. After reading the experiments, my questions are

- it is good to compare with baseline methods, but are these baseline methods the state-of-the-art?

- There is extensive study on the UrbanSound8K data. How does the performance of wavelet network compare to a broader range of methods beyond the CNN backbone? In particular, transformers are powerful network architectures for sequential data processing; are wavelet networks comparable to transformers?

**Strengths And Weaknesses:**

--------------- Strength:

The paper devotes many introductory efforts to distinguishing visual and auditory data, where the latter can be represented in the 2D time-frequency plane. This contradictory exposure highlights the difficulty of learning with auditory data and elucidates the insufficiency of standard CNNs.

Derivations and definitions regarding groups and group convolutions are clearly stated with many graphical illustrations. This provides an easier understanding towards the complicated math expressions.

Experimental results support the efficiency of the wavelet network, as it uses less parameters with good performance.

--------------- Weakness:

There is no obvious weakness, but please refer to the requested changes for a detailed discussion.

---

> ### Author Response · Authors · 2023-10-18
>
> Dear reviewer A2P8,
>
> Thank you very much for review. We sincerely appreciate the time you spend in evaluating our work.
>
> Here, we respond to your observations:
>
> * **Section 4 might need a different highlighting style. Currently, the definition of Fourier transform and wavelet transform...**
>
> You are completely right about the purpose of the section, and we completely understand your concern. Our idea with the gray boxes was to provide context to the readers without breaking the flow of the text. But, we realize now that we did not state what the purpose of the gray boxes were anywhere. Do you think it would be enough to have a clarification regarding their purpose at the beginning of this section? Do you have any other ideas that could help highlight the purpose of the section?
>
> * **Section 5 looks very complicated. It begins with definitions of group convolutions, ...**
>
> The current structure of this section is as follows:
>
> ---
> 5. Wavelet networks: Scale-translation equivariant learning from raw waveforms
>
>    5.1. Scale-translation preserving mappings: group convolutions on the scale-translation group
>
>    5.2 Wavelet Networks: architecture and practical implementation
>
>          5.2.1 Group convolutional kernels on continuous bases
>
>          5.2.2 Constructing a discrete scale grid
>
>          5.2.3 Imposing wavelet structure to the learned convolutional kernels
>
>    5.3 Wavelet networks perform nested non-linear time-frequency transforms
> ---
>
> Am I correct in saying that the structure you propose is as follows?
>
> ---
> 5. Wavelet networks: Scale-translation equivariant learning from raw waveforms
>
>    5.1 Wavelet Networks: architecture
>
>    5.2 Scale-translation preserving mappings: group convolutions on the scale-translation group
>
>    5.3 Wavelet networks perform nested non-linear time-frequency transforms
>
> 6 Implementation Details
>
>      6.1 Group convolutional kernels on continuous bases
>
>       6.2 Constructing a discrete scale grid
>
>       6.3 Imposing wavelet structure to the learned convolutional kernels
> ---
>
> I believe this alternative could also work out well. But I feel that the current structure is better in that it provides the necessary theory, then delves into its practical implementation and ends with an analysis on the modus operandi of the resulting architecture. With that being said, we are happy to change the structure of this section if you feel that the changes simplify the structure of the document.
>
> * **it is good to compare with baseline methods, but are these baseline methods the state-of-the-art?**
>
> To the best of our understanding, these methods are the SotA on end-to-end learning, and are at least comparable with the state-of-the-art in the general sense. It is true that newer methods exist, but these rely on preprocessing, e.g., through Mel-Spectrograms [1], or make use of self-attention blocks [2]. This is why we did not consider them as backbones in our approach.
>
> * **How does the performance of wavelet network compare to a broader range of methods beyond the CNN backbone? In particular, transformers are powerful network architectures for sequential data processing; are wavelet networks comparable to transformers?**
>
> There has been research involving how to incorporate equivariance in self-attention architectures [3]. However, the problem is that this rapidly becomes expensive due to the quadratic cost of self-attention. Although this might become feasible in the future, we did not consider methods that are not purely based on convolutions, to avoid incurring in additional costs, which are already larger than that of normal CNNs through the use of parallel convolutions with dilated kernels --Sec. 7--. In addition, there have been several advances in sequence modeling, which seem to pinpoint that convolutional methods may be just as powerful in a more efficient manner [4, 5].
>
> **References**
>
> [1] Dentamaro, Vincenzo, et al. "AUCO ResNet: an end-to-end network for Covid-19 pre-screening from cough and breath." Pattern Recognition 127 (2022): 108656.
>
> [2] Gazneli, Avi, et al. "End-to-end audio strikes back: Boosting augmentations towards an efficient audio classification network." arXiv preprint arXiv:2204.11479 (2022).
>
> [3] Romero, David W., and Jean-Baptiste Cordonnier. "Group equivariant stand-alone self-attention for vision." arXiv preprint arXiv:2010.00977 (2020).
>
> [4] Gu, Albert, Karan Goel, and Christopher Ré. "Efficiently modeling long sequences with structured state spaces." arXiv preprint arXiv:2111.00396 (2021).
>
> [5] Poli, Michael, et al. "Hyena hierarchy: Towards larger convolutional language models." arXiv preprint arXiv:2302.10866 (2023).
>
> ---
>
> We hope that these responses clarify your questions and concerns. Please let us know if you have any follow-up / additional questions.
>
> Best regards,
>
> The Authors

---

### Review · Reviewer_WoBG · 2023-11-21

**Summary Of Contributions:**

In 2D and 3D data domain, prior works have proposed to exploit equivariances to different symmetry groups to improve generalization performance, and parameter and data efficiency. However, in the 1D domain, this has not been explored much beyond the usual translation group exploited by conventional CNNs.

The paper proposes Wavelet networks to learn from time series data by exploiting equivariance to the scale-translation group. The convolutional layers for this group are constructed following the general formulation of group convolution proposed in [1] and seems somewhat similar to the prior scale-translation equivariance work [2] proposed for the 2D domain.

To avoid interpolation while lifting to different scales, convolutions are performed in the continuous domain by parameterizing the convolutional kernels using splines and the scale grid is suitably discretized to approximate the convolution integrals.

The paper also highlights the similarities between the scale-translation group convolution and the definition of wavelet transform. To promote a wavelet-like structure in the learnt convolutional kernels, they are regularized during training to have zero mean. The authors show that this interestingly provides a consistent improvement in performance.

Experimental results show that the proposed wavelet networks improve performance over baseline models on different learning tasks.

[1] Cohen, Taco, and Max Welling. "Group equivariant convolutional networks." International conference on machine learning. PMLR, 2016.

[2] Sosnovik, Ivan, Michał Szmaja, and Arnold Smeulders. "Scale-Equivariant Steerable Networks." International Conference on Learning Representations. 2019.

**Audience:**

No

**Claims And Evidence:**

Yes

**Requested Changes:**

Please refer to my comments in the Strengths And Weaknesses section

**Strengths And Weaknesses:**

**Strengths:**
1. Barring a few minor typos, the paper is well written, and the text and illustrations are generally easy to follow.

   I particularly like the discussion in section 4 about the limitations of using convolutional neural networks over spectro-temporal data. Inspired by the success of CNNs on images, many prior works convert time series into 2D domain by computing spectrogram (or some variant) and pass them to a 2D CNN. The authors highlight why this is not the best strategy since unlike images, spectrograms of time series are often highly non-local.

2. The noted improvement in performance obtained by imposing a ‘wavelet-like’ structure on the filters is interesting.

**Weaknesses:**

While I like the way this paper is written and the experimental results are also nice, I currently have one major concern that I would like the authors to clarify.

It is not clear to me what the key technical contribution is, in this paper. The authors mention that the main motivation is to build scale-translation equivariant networks for 1D data. However, is there any particular challenge in the 1D case that makes it non-trivial to directly extend the prior 2D scale-translation equivariant approaches to it? It appears to me that the proposed approach is quite similar to the 2D case of Sosnovik et al. [2]

---

> ### Author Response · Authors · 2023-11-28
>
> Dear reviewer R4e9,
>
> Thank you very much for review. We sincerely appreciate the time you spend in evaluating our work.
>
> Here, we respond to your observations:
>
> **It is not clear to me what the key technical contribution is, in this paper.**
>
> +-> As specified at the end of Section 1, the core of our contribution is the exploration of equivariant learning on time-series data. Based on our analysis, we find translation and scale to be important for time-series processing and explore this idea thoroughly --which to the best of our knowledge has not been done previously. As such, we do not aim to construct a "better scale-equivariant model" than that of [2], but rather to explore the use of symmetry preservation by means of scale-translation equivariance for time-series processing.
>
> With that being said, It is true that existing equviariance works, e.g., Sosnovik et al. (2019), Worrall & Welling (2019), could be extended to the 1 dimensional case. However, it is important to note that the scales on which 1D data works is very different that those observed for images, and therefore, it is not possible to transfer insights gained in 2D to 1D directly. Specifically, the largest images considered in Sosnovik et al. had size 96x96, whereas the audio samples that we use easily go beyond 32k length. This has important implications for the way in the scales in the scale-equivariant model interact with one another. For example, Sosnovik et al. state that `"...we see that using SESN with interscale interaction introduces extra equivariance error due to the truncation of S. We will build the networks with either no scale interaction or interaction of 2 scales..."`, which is very different from what we observe in 1D (see Tabs 4-6). In our case we observe that interscale interactions play an important role, and in fact, we observed that considering interscale interactions among more scales led to better results. In our case, the number of scales considered are rather constrained by the size of the convolutional kernels used and their cost (Sec. 5.2.2).
>
> If the reviewer agrees, we can extend on this in the discussion after the experimental section to make this explicit.
>
> ---
>
> We hope that our response clarify your concern. Please let us know if you have any follow-up / additional questions.
>
> Best regards,
>
> The Authors

---

> ### Comment · Reviewer_WoBG · 2023-12-06
> **Responding to authors' comment**
>
> I thank the authors for their response.
>
> Their point on inter-scale interactions is interesting and I think a discussion on how time series data relates to it in comparison to image data from [2] would be a helpful addition to the paper.

---

### Decision · Action_Editor_EuJS · 2023-12-06

**Recommendation:** Accept as is

**Comment:**

Based on my comments above, I am recommending this paper is accepted as is. I would recommend the authors do a complete proof-read of the paper to prepare their camera-ready version (there are still some minor typos, see e.g. sentence prior to Eq.17).

**Audience:**

This manuscript will certainly be of interest to researchers working on ML applications for audio, finance, and other areas that rely on the analysis of time-series.

**Claims And Evidence:**

This paper present a construction of wavelet networks, which provide parametric models for learning scale and translation equivariant functions from real data, and which are therefore particularly well-suited for time-series. Unlike most works that seek to enforce equivariances for 1D signals, the work presented here provides strict equivariance to scale and shifts while comparing to the resulting maps of wavelet tarnsforms. The manuscript is very clear, the background is comprehensive and valuable, and their experimental results are compelling and satisfactory.

---

> ### Author Response · Authors · 2023-12-22
>
> This is great news.
>
> We deeply appreciate your time and that of the reviewers invested in reviewing our work. We will shortly update the paper following the last comments of the reviewers as well as your recommendations.
>
> Best,
>
> The Authors.